# Oxidation-boosted charge trapping in ultra-sensitive van der Waals materials for artificial synaptic features

Feng-Shou Yang[1,2,9], Mengjiao Li [1,9 ✉], Mu-Pai Lee[1,3], I-Ying Ho[1,2], Jiann-Yeu Chen[4], Haifeng Ling[5], Yuanzhe Li[5], Jen-Kuei Chang[1], Shih-Hsien Yang[1], Yuan-Ming Chang[1], Ko-Chun Lee[1], Yi-Chia Chou [3], Ching-Hwa Ho [6], Wenwu Li [1,7 ✉], Chen-Hsin Lien[2 ✉] & Yen-Fu Lin [1,4,8 ✉]

Exploitation of the oxidation behaviour in an environmentally sensitive semiconductor is significant to modulate its electronic properties and develop unique applications. Here, we demonstrate a native oxidation-inspired InSe field-effect transistor as an artificial synapse in device level that benefits from the boosted charge trapping under ambient conditions. A thin $InO_x$ layer is confirmed under the InSe channel, which can serve as an effective charge trapping layer for information storage. The dynamic characteristic measurement is further performed to reveal the corresponding uniform charge trapping and releasing process, which coincides with its surface-effect-governed carrier fluctuations. As a result, the oxide-decorated InSe device exhibits nonvolatile memory characteristics with flexible programming/erasing operations. Furthermore, an InSe-based artificial synapse is implemented to emulate the essential synaptic functions. The pattern recognition capability of the designed artificial neural network is believed to provide an excellent paradigm for ultra-sensitive van der Waals materials to develop electric-modulated neuromorphic computation architectures.

[1] Department of Physics, National Chung Hsing University, Taichung 40227, Taiwan. [2] Department of Electrical Engineering and Institute of Electronic Engineering, National Tsing Hua University, Hsinchu 300, Taiwan. [3] Department of Electrophysics, National Chiao Tung University, Hsinchu 300, Taiwan. [4] Research center for Sustainable energy and Nanotechnology, National Chung Hsing University, Taichung 40227, Taiwan. [5] Institutes of Advanced Materials (IAM), Nanjing University of Posts&Telecommunications, Nanjing 210023, China. [6] Graduate Institute of Applied Science and Technology, National Taiwan University of Science and Technology, Taipei 106, Taiwan. [7] Engineering Research Center for Nanophotonics and Advanced Instrument (MOE), Technical Center for Multifunctional Magneto-Optical Spectroscopy (Shanghai), School of Physics and Electronic Science, East China Normal University, Shanghai 200241, China. [8] Institutes of Nanoscience, National Chung Hsing University, Taichung 40227, Taiwan. [9]These authors contributed equally: Feng-Shou Yang, Mengjiao Li. ✉email: mengjiaoli@email.nchu.edu.tw; wwli@ee.ecnu.edu.cn; chlien@ee.nthu.edu.tw; yenfulin@nchu.edu.tw

The sustainable increase in the number of integrated basic components on a chip is becoming a significant challenge for the advancement of the internet of things (IoT) technology, especially in the post-Moore era. In terms of the scaling down limitation of silicon-based materials, two-dimensional van der Waals semiconductors have aroused widespread interest for the next generation of nanoelectronics owing to the atomic-scale thickness, large area development, and good charge transport behaviour[1–6]. These attractive characteristics endow van der Waals semiconductor-based electronics with an immunity to short-channel effects, gate controllability, or tunable polarity of conductivity, which facilitate promising potential uses in transistors, sensors, and memory[7–12]. Nevertheless, despite the abovementioned advantages, compared with traditional silicon materials, most members of the environmentally sensitive semiconductor family encounter very noticeable oxidation upon exposure to ambient atmosphere. The representative examples are black phosphorus (BP), indium selenide (InSe), hafnium sulfide (HfS$_2$), molybdenum ditelluride (MoTe$_2$), and the layered organics like Ruddlesden–Popper perovskites (RPP), which exhibit discernible surface morphological variations even over a few minutes[7,12–18]. In addition, the carrier transport in devices based on these semiconductors would be randomly disturbed and lead to severe degradation effects in electrical or optical performance, which are acknowledged as the most critical obstacles for their successful practical application. To address such a poor stability of layered semiconductors, several protection strategies have been introduced into the structural design to achieve effective passivation, such as encapsulation methods, surface doping, interfacial engineering, and ionic liquids[19–21]. However, achieving both the long-term stability and excellent device performance is still challenging; therefore, a determination as to how to reasonably use the unavoidable oxides in environmentally sensitive semiconductors seems to be another alternative to control the inner charge transport characteristics and is meaningful for developing both charge trapping memory and artificial synaptic systems. A synapse, which is the basic building block of a neural network in human brains, functions as the junction between the pre-neuron and post-neuron to realize information transmission, learning, and storage[22–25]. Various orders, such as emotions and actions, can be solved in parallel by a neural network under ultra-low power consumption and fault tolerance, which is considered as the natural paradigm for the innovation of brain-inspired neuromorphic computing technology in the modern information industry because of the physical limitation of separated memory and central processor in the classic von-Neumann computation architecture[26–29]. Previously, tens of silicon-based transistors were employed to mimic the synaptic functions, which was hindered by the circuit complexity and huge energy consumption for large-scale integrations[30,31]. Ionic-gating-modulated transistors were recently proposed to emulate the information transmission in biological synapses based on the reversible electrochemical proton doping process, although the device repeatability remains out of control[32,33]. In this regard, the use of a sensitive van der Waals materials-based transistor, which will benefit from its native oxidation effect and tunable charge trapping behaviour, is believed to possess innate advantages to achieve design-simplified synaptic devices[34]. Therefore, there is more motivation to conduct an in-depth investigation of the charge trapping mechanism in environmentally sensitive van der Waals materials for mimicking operation-controllable artificial synapses.

In this work, we experimentally demonstrate an oxidation-boosted artificial synaptic device, for the first time, based on a typical sensitive van der Waals semiconductor, InSe. As a representative of the III-VI group materials, InSe possesses small effective electron mass and excellent intrinsic charge transport characteristics. Its acknowledged air-instability and large surface to volume ratio, resulting in severe performance degradation and hysteretic behaviour, attract much researchers' attention on how to protect it from contact with air for steady electrical properties, including our team. To put this in perspective, the environmental sensitivity provides itself favourable conditions for introducing into oxidation layer; thus quantitatively analyzing and comprehending carrier traps in this thin oxide layer is of great significance in particular for van der Waals materials to further limit its negative impact, even utilize it for enhancing the performance and applications of electronic devices[14,35–40]. Intriguingly, the microscopic structure observation of InSe FET has confirmed that the native oxide formed at the bottom of the InSe channel under ambient conditions is capable of serving as a unique charge trapping layer to modulate the charge transport behaviour. The dynamic charge trapping and releasing process is visually delineated by low-frequency noise analysis according to the surface effect dominated carrier fluctuations, which is responsible for its nonvolatile data storage characteristics with reliable programming/erasing operations. Furthermore, the basic synaptic functions, which include short-term plasticity (STP) of paired-pulse faciliation (PPF) and the long-term plasticity (LTP) of spike-timing-dependent plasticity (STDP), are successfully mimicked by this oxidation-inspired InSe artificial synaptic device, as well as the system-level pattern recognition based on the artificial neural network (ANN). Consequently, the ingenious use of the oxidation layer in the InSe FET offers a new opportunity for air-unstable layered semiconductors to bulid a simpler configuration in the field of data storage and next-generation neuromorphic computation.

## Results

**Characteristics of InSe FET under different conditions.** Here, the demonstration of the InSe artificial synaptic features in device level was inspired by the specific electrical behaviour of the surface-doped InSe FET under an ambient atmosphere, and its schematic structure is shown in Fig. 1a. Note that the electrical characteristics for InSe FETs with/ without surface doping effect were provided in Supplementary Fig. 1 for comparison[38]. The exfoliated InSe flake was intentionally selected as 10–20 layers under optical microscopy (OM) for improved device properties[41]. Its typical thickness was examined to be around 13.8 nm by the topographic scan profile of atomic force microscopy (AFM) (Fig. 1b). In consideration of the ultrahigh sensitivity to moisture and oxygen for InSe, first, we characterized the basic electrical performance of the fabricated InSe FET to explore the underlying charge transfer differences between ambient (InSe A-FET) and vacuum (InSe V-FET) conditions. All of the statistical transfer curves and the gate leakage in Fig. 1c and d for 15 representative InSe FETs under two conditions indicated the typical electron-conduction behaviour in the InSe channel. Along the sweep directions of gate voltage varying between −80 and +80 V, the recorded transfer curves for InSe V-FET exhibited weak electrical hysteresis loops. Such a slight positive shift of the threshold voltages below 20 V in backward gate sweeping can be attributed to the charge transfer effect instead of capacitive gating effect due to the few interface states between the channel and dielectric layer or fabrication-induced defects[42–46]. While immersing the same 15 devices in ambient conditions, the observed transport windows were consistently enlarged, which indicated the boosted electrical hysteresis phenomena. Figure 1e shows the corresponding statistical threshold voltage shift under different conditions based on 25 InSe FETs. Its average value in the InSe A-FET approached 100 V, hinting at the existence of unique charge trapping and

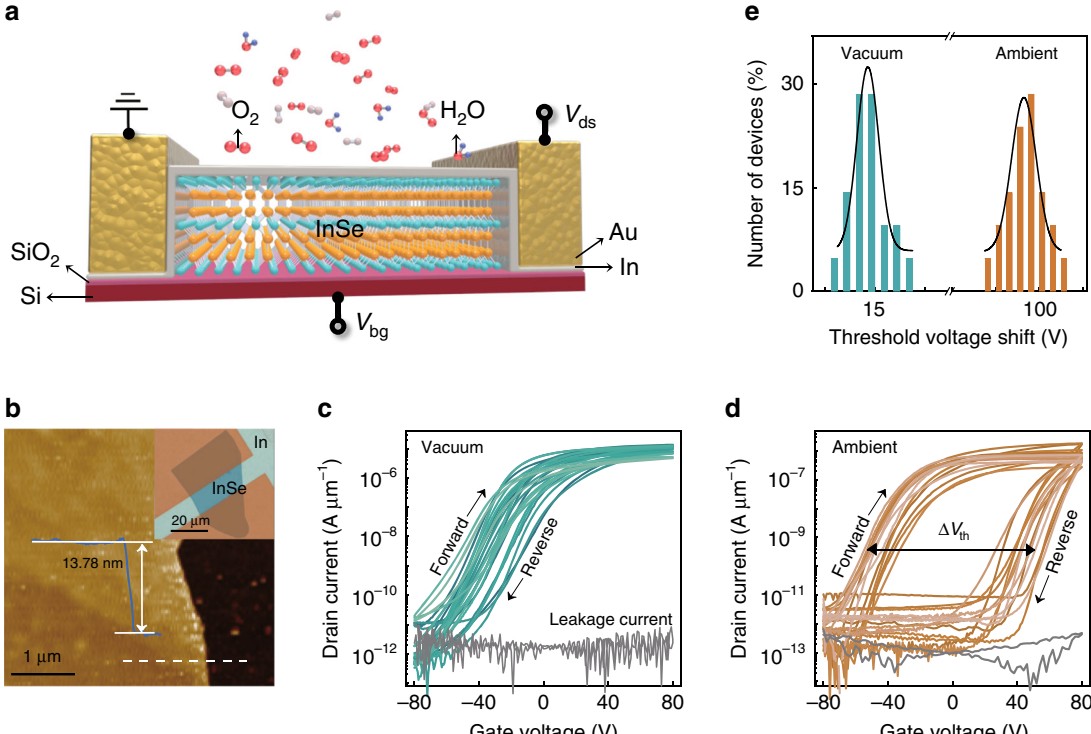

**Fig. 1 Structure schematic and basic electrical properties of InSe FET. a** Schematic diagram of the InSe FET exposed to ambient conditions, in the presence of several gas molecules (such as $H_2O$ or $O_2$). The Si substrate serves as the gate electrode and the covered In film acts as the surface dopant. **b** AFM image and the corresponding height profile of the InSe channel. The inset shows the typical OM image of the InSe FET, where the length and width of the InSe channel are 13 and 31 µm, respectively. **c, d** Transfer characteristics and the gate leakage curves at drain voltage $V_{ds} = 0.1$ V for the 15 representative InSe FETs under the vacuum and ambient conditions, respectively. Note that the arrows mark the directions of the gate voltage sweep and $\Delta V_{th}$ means the threshold voltage shift. **e** Histogram of the threshold voltage shift for InSe V- and A-FETs and the Gaussian fit (solid lines) based on the statistical database.

releasing events in the InSe channel owing to the possible native oxidation of InSe under ambient conditions, which will not occur under vacuum conditions. To preliminary verify the hypothesis of the native oxidation event, the specific experiments were intentionally designed to detect the variation of the transfer curves for InSe FET depending on the exposing time and immersing conditions[47]. On the one hand, Supplementary Fig. 2 shows the evolution of transport windows along with the exposing time under ambient conditions. Initially, a narrow window in the transfer curve for InSe FET can be observed under vacuum conditions. With the increase of the exposing time under ambient conditions, the obtained hysteresis loop gradually enlarged and then saturated with a window of about 90 V. Moreover, such a wide hysteresis loop under ambient conditions over 2 h can still be well-retained without obvious variation, even when storing it back to vacuum conditions again, which minimizes the impact of the physical absorption of gas molecules[48]. On the other hand, when storing a fresh sample in the nitrogen ($N_2$) or argon ($A_r$) atmosphere for around 30 minutes, the measured transport windows are almost fully consistent with that of InSe V-FET (Supplementary Fig. 3). While in the case of dry air, it presents an enlarged hysteresis loop, even being stored back to the vacuum conditions, suggesting the weak influence of the hydrolysis effect on the InSe channel. In consequence, it is believed that the enhanced hysteretic behaviours in InSe A-FET are dominated by the native oxide layer under ambient conditions. It is worth noting that the excellent electrical properties of InSe A-FET, which include high on current, on-off current ratio, and good Ohmic contact (see the corresponding output curves in Supplementary Fig. 4), were maintained compared with those in InSe V-

FET, without obvious electrical degradation. In terms of the above, such an intriguing electrical hysteresis behaviour in the InSe A-FET not only motivates more efforts in exploring the underlying charge trapping mechanisms but also suggests its potential use in memory systems[49,50].

For visually examining the occurrence of the native oxidation event, we performed the microscopic structure observation of the InSe A-FET using high-resolution-transmission electron microscopy (HRTEM). The cross-sectional image and the corresponding energy-dispersive X-ray spectroscopy (EDS) element mapping distinctly present the components of the device are shown in Fig. 2a, which include the Au contact electrode, In doping layer, InSe channel, and $SiO_2$ dielectric layer. The highly crystalline lattice of the InSe channel was approximately 16 layers, which agreed well with the height profile obtained from AFM analysis. Intriguingly, a thin amorphous layer inserted between the InSe and $SiO_2$ was evidently observed. We then conducted EDS line scan analysis along the cross-section region of InSe A-FET to identify the composition and thickness of this amorphous layer. The profiles of In, Se, and Si elements in Fig. 2b clearly outlined the interfacial region between InSe and $SiO_2$. In particular, the intensities of elemental O could be ignored at the region of InSe layer while they underwent a sharp increase at the top side of $SiO_2$, which signified the existence of an amorphous 2-nm-thick $InO_x$ layer at the interfacial region. It is worth mentioning that the thin $InO_x$ layer was not detected for the devices of InSe V-FET[38]. Therefore, the native $InO_x$ formed under ambient environment was believed to be a vital layer to boost the charge trapping and releasing behaviour between the InSe channel and $InO_x$ layer to result in the enlarged transport window in transfer

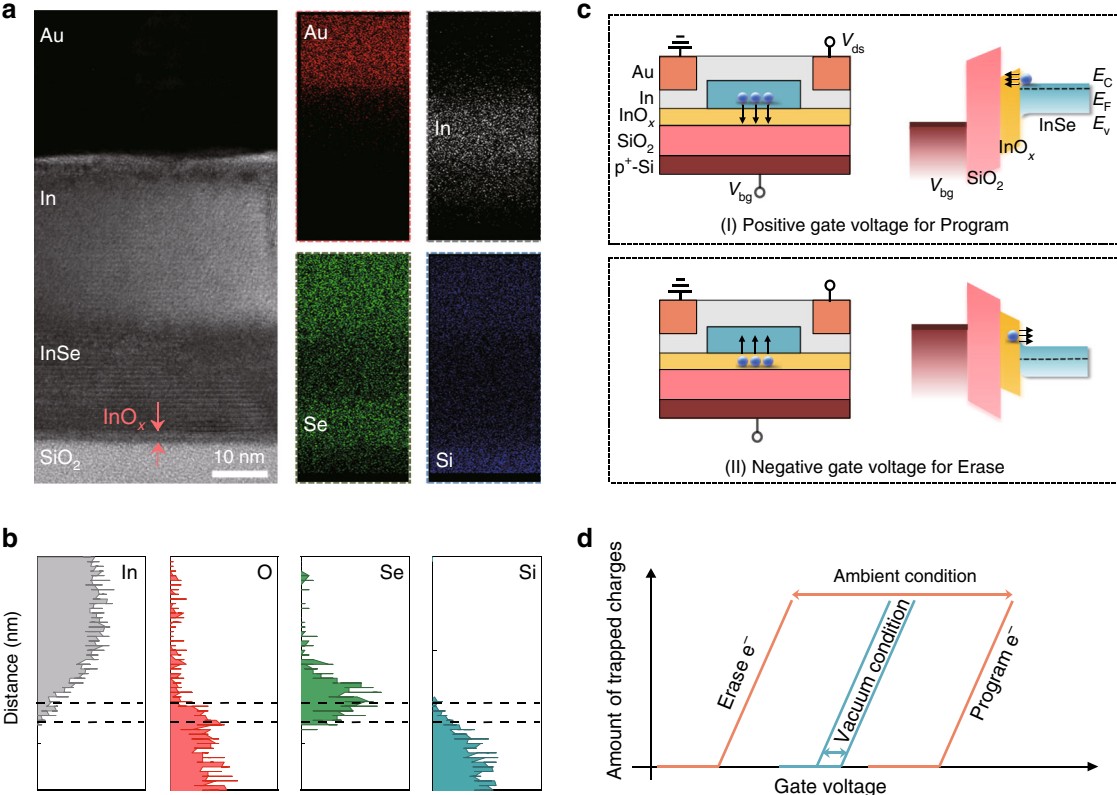

**Fig. 2 Observation of the native oxidation layer by microscopic structure analysis. a** Cross-sectional HRTEM image and the corresponding EDS element mappings of a typical InSe A-FET. The arrows highlight the existence of the $InO_x$ layer. **b** EDS line scan profiles of In, O, Se, and Si elements along the cross-section region of the device. Note that the dash lines label the region of $InO_x$. **c** Energy band structure of the charge trapping and releasing process of Program and Erase operations for InSe A-FET under different gate voltages ($V_{bg}$). The Program and Erase operations are defined as the charge trapping process by $InO_x$ and charge releasing process from $InO_x$, respectively. **d** Schematic diagram of boosted threshold voltage variation for InSe A-FET compared with that for InSe V-FET.

characteristics. Then, we determined the energy band and the corresponding operation states under the modulation of gate voltage to better understand the charge trapping and releasing process in InSe A-FET. The (I) part of Fig. 2c defines the Program operation by applying a positive gate voltage. In this process, the electrons in InSe channel were trapped by the $InO_x$ layer, which leads to a positive shift of $V_{th}$ and the low current level in InSe A-FET. While under a negative gate voltage (the (II) part of Fig. 2c), the trapped electrons were released from $InO_x$ because of the tilted energy of the trapping layer on another side, thus bringing about the threshold voltage shift towards a negative gate voltage and triggering the Erase operation in InSe A-FET with a high current level. Therefore, one can conclude that the charge trapping behaviour in InSe FET have been enhanced under ambient condition, compared with that of vacuum condition, and benefits from the native oxidation (Fig. 2f), which provides an ideal configuration to extend the promising opportunity in the field of charge trapping-based memory as well as artificial synaptic devices.

To understand the mechanism of the electric fluctuations versus device signal for building sensitive materials-based electronics toward practical applications, the systematical dynamic characteristic measurement for InSe FETs under different conditions was conducted[51,52]. The noise power spectra map of current fluctuations ($S_I$) for InSe A- and V-FET as the frequency ranged from 2 to 10 kHz are shown in Fig. 3a and Supplementary Fig. 5. With increasing the applied gate voltage, the measured $S_I$ was incrementally augmented, and exhibited the high consistency of the ideal $1/f$ signal (a representative curve

below the map). Figure 3b profile the drain current normalized $S_I$ demonstrated its independent characteristic of drain voltage, which revealed that the electric noise signal was largely driven by the InSe conducting channel instead of the contact barrier effect between the channel and source/drain electrodes. This phenomenon observed from InSe A-FET was attributed to the In dopant induced good Ohmic contact behaviour because of the lower work function of In metal than that of Au electrode[43]. In detail, the degree of the $1/f$ dependence in the measured noise power spectra was determined based on the empirical formula $S_I \propto I_{ds}^{\alpha}/f^{\beta}$, where $\alpha$ and $\beta$ represent the scaling exponents for the current and frequency, respectively. As provided in Fig. 3c, the derived values of $\beta$ were approximately 1.36 ± 0.01 and 1.21 ± 0.01 for InSe A- and V-FET, respectively. This suggested a uniform distribution of charge trapping or scattering states both in space and energy under the two conditions. Furthermore, both the fitted values of the exponent $\alpha$ for the ambient and vacuum conditions were near to 2, which revealed that the $1/f$ electric noise behaviour in InSe FETs could be assigned as the resistor fluctuation[53].

Considering that the extracted scaling exponents showed high similarity between InSe A- and V-FET, we further discuss the nature of the $1/f$ characteristics under the two conditions by combining the carrier number fluctuation and correlated mobility fluctuation in terms of the dynamic carrier trapping behaviour and the surface scattering[54–56]. The detailed descriptions are provided in the Supplementary Note 1. To discriminate the governing mechanisms of the noise source in the InSe devices under different environments, Fig. 3d plots the log-log scaled

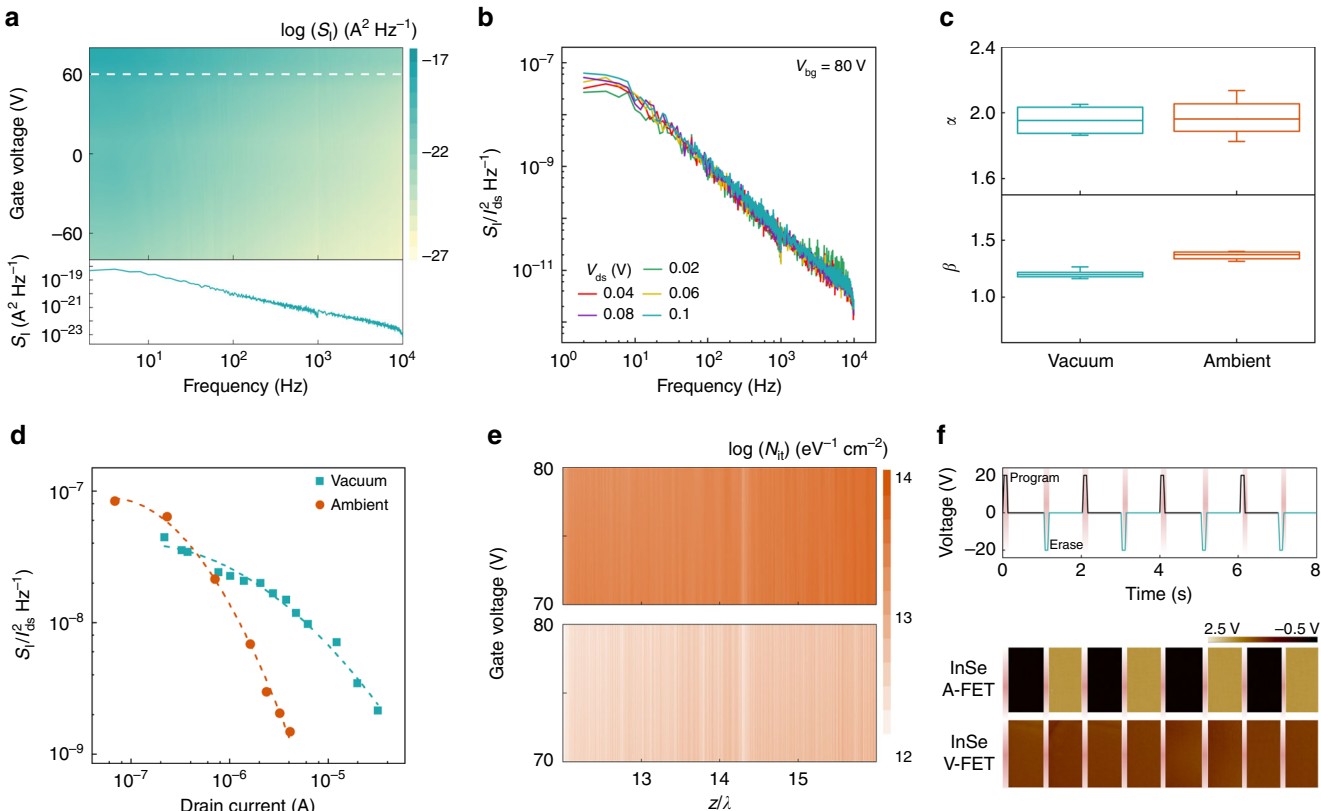

**Fig. 3 Governing mechanisms of the carrier transport in InSe FET. a** Recorded $S_I$ as a function of frequency and gate voltages for InSe A-FET. Note that a $S_I$ curve at $V_{bg} = 60$ V based on the white dash line (upper) is profiled a representative of $1/f$ noise (under). **b** Drain current normalized $S_I$ depending on $I_{ds}$ for InSe A-FET at several $V_{ds}$. **c** Box-plots of the fitted $\alpha$ (upper) and $\beta$ (lower) values for InSe V- and A-FET. Note that the center line, the upper, and the lower bars correspond to the average, the maximum, and the minimum values, respectively. **d** Normalized $S_I$ by $I_{ds}^2$ (discrete dots) versus $I_{ds}$ on a log-log scale for InSe FET under different conditions. The dashed lines are the well-fitted results based on the correlated mobility fluctuation. **e** The extracted values of effective interface trap density for InSe V- and A-FET as a function of $z/\lambda$ and gate bias. **f** The dynamic variations of the surface potential (under read states) of InSe channel by in situ KPFM depending on the scanning time, for InSe A- and V-FET, respectively. The program and erase operations were defined by applying a +20 V and a −20 V gate voltage pulse (5 s) for 4 cycles.

$S_I/I_{ds}^2$ as a function of drain current. The evolution of $S_I/I_{ds}^2$ for InSe A-FET tended to depend on the change of $I_{ds}^{-2}$, which indicated that the surface effect dominated the carrier transport. This phenomenon validated the existence of charge trapping/releasing events at the InSe channel and dielectric interface, which was high-consistency with the HRTEM analysis. However, the observed $S_I/I_{ds}^2$ for InSe V-FET is inclined to mainly obey the variation of $I_{ds}^{-1}$. The corresponding $\alpha_{SC}$, a mobility fluctuation related Coulomb scattering coefficient, was fitted to be $2.69 \times 10^4$ V s C$^{-1}$ for InSe V-FET, which was ~100 times higher than that for InSe A-FET ($2.95 \times 10^2$ V s C$^{-1}$), indicating the bulk condition mainly governed carrier transport and fluctuations in InSe V-FET. Furthermore, the effective trap densities, $N_{it}$, which is composed of all the trap states from both the bottom interface and the top surface of the InSe channel in the present device configurations, were evaluated to visualize the process of charge transport in InSe FETs under the two conditions[57]. For a better comparison, the extracted values of $N_{it}$ as a function of $z/\lambda$ and gate voltages under two conditions are shown in Fig. 3e, where $z$ and $\lambda$ represent the trap depth and tunnelling distance parameter, respectively[51]. The $N_{it}$ values were independent of $z/\lambda$ under each applied gate voltage, which further portrayed the observed in the $1/f$ behaviour. The overall value of $N_{it}$ for InSe A-FET was near $10^{14}$ eV$^{-1}$ cm$^{-2}$, at least an order of magnitude at least higher than that in the vacuum condition. Such a big difference of $N_{it}$ values between the two conditions is believed to make the boosted

charge trapping/detrapping events between the InO$_x$ interfacial layer and InSe channel in InSe A-FET, shedding more light on the evident surface effect under ambient conditions because of the formation of 2-nm-thick native InO$_x$ layer. On the other hand, we further conducted the in situ KPFM measurements to visually verify the charge trapping effect of InO$_x$ via examining the dynamic variation of the surface potential of InSe channel in InSe A- and V-FET (Fig. 3f). In the case of InSe A-FET, the recorded potential of InSe channel exhibits distinct difference under the read states ($V_{bg} = 0$ V) after the program and erase pulses, as well as a good reproducibility in 4 successive cycles, which shed light on the occurrence of the charge trapping/detrapping events in InSe A-FET during the program/erase process. In contrast, the potential difference in InSe V-FET could almost be ignored under several read states, hinting at its weak charge transfer behaviour. Thus we could determine the dominant role of the InO$_x$ layer in trapping/detrapping electrons in the proposed devices, which paves the way for the demonstration of memory and artificial synaptic features.

**The nonvolatile memory properties bsaed on InSe A-FET.** The clear charge transport mechanisms in InSe A-FET revealed by the structure and dynamic characteristics analysis inspired more exploration of charge trapping-based memory systems. Figure 4a shows the 3D schematic of InSe A-FET, in which the thin InO$_x$ layer formed under the bottom of the InSe channel provided the

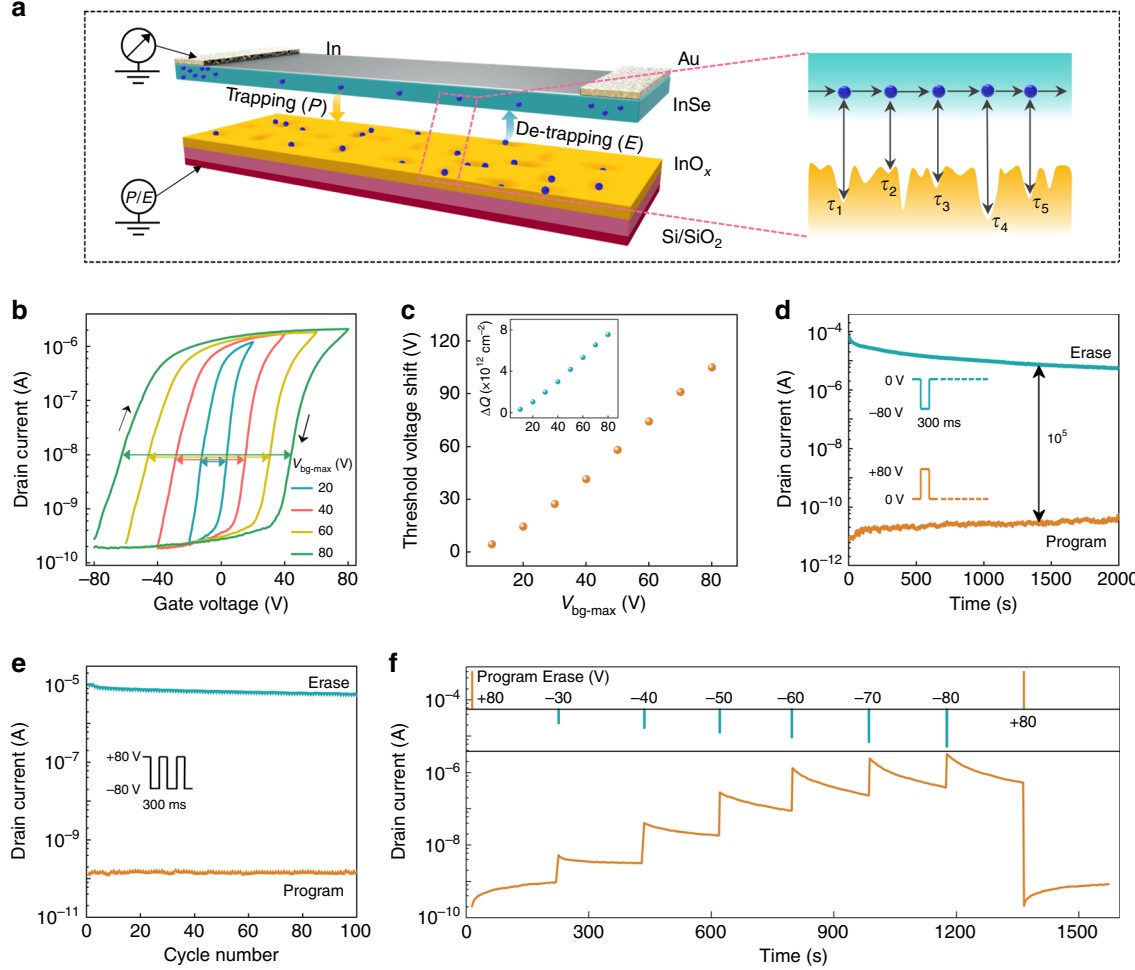

**Fig. 4 Performance characterizations of the InSe A-FET based memory device. a** 3D schematic of InSe A-FET with the existence of $InO_x$ to achieve the charge trapping/ de-trapping under applied operation voltage. The inset on the right side delineates the uniform distribution of trapping states for the dynamic fluctuation process between the InSe channel and $InO_x$ layer. Note that $\tau$ means the time constant for different traps. **b** Transfer characteristics of InSe A-FET with the $V_{bg\text{-}max}$ (the maximum value of the swept range of gate voltage). The drain voltage is set as 0.1 V and the colourful arrows mark the increasing threshold voltage shift. Note that the arrows indicate the directions of hysteresis loops. **c** Variation of the threshold voltage shift depending on $V_{bg\text{-}max}$. Inset shows the extracted amounts of trapped charges under various $V_{bg\text{-}max}$. **d** Retention performance of InSe A-FET based memory. The currents were separately read under Program and Erase states with $V_{ds} = 0.1$ V and $V_{bg} = 0$ V. **e** Endurance characteristics of the memory device. Program and Erase operations were carried out under cyclic voltage pulses. The Read state was operated under $V_{ds} = 0.1$ V. **f** Evolution of read current of the memory device under various operation states, which indicates its multiple data storage characteristics.

innate situation for demonstrating memory operations. Accordingly, a concrete process of the dynamic charge trapping and releasing events between the InSe channel and $InO_x$ layer is delineated in the right illustration. To gain insight into the charge retention ability of the $InO_x$ layer, to the best of our knowledge, this is the first basic performance evaluation of InSe A-FET based memory. The transfer characteristics of InSe A-FET under consecutive voltage sweeps are illustrated in Fig. 4b. As marked by the arrows, a bigger $V_{bg,\ max}$ resulted in a wider hysteresis window. The extracted values of threshold voltage shift in Fig. 4c were positively proportional to the applied $V_{bg,\ max}$, which indicated that the charge trapping behaviours in InSe A-FET could be effectively modulated by varying the sweep range of gate voltage. Furthermore, the amounts of trapped charges ($\Delta Q$) as a function of $V_{bg}$ were estimated to be around $10^{-12}$ cm$^{-2}$ (inset of Fig. 4c) based on the capacitance model of $\Delta Q = \Delta V_{th}\ C_{ox}/q$, where $C_{ox}$ and $q$ are the capacitance of a 300-nm SiO$_2$ dielectric layer and the elementary charge, respectively[58]. This result was slightly smaller than the difference value of $N_{it}$ between InSe A- and V-FET (Fig. 3f), where the underestimated trapping sites here

possibly stemmed from both the non-saturated states under current Program/Erase operations and additional origins, such as the fabrication-induced defects or interface states between the channel and dielectric layer.

Generally, excellent charge retention ability and durability are of great significance to the practical use of energy-efficient memory systems. Figure 4d shows the reliable retention performance of InSe A-FET based memory with a high Program/Erase ratio above $10^5$, for which the read currents exhibited negligible degradation even after $2 \times 10^3$ s. Note that the Program and Erase states here were defined by the application of a positive and a negative gate voltage pulse, respectively, to excite the initial charge trapping states in InSe A-FET. Besides, the two distinctive Program and Erase states could be well-maintained over $10^4$ s without obvious degradation (Supplementary Fig. 6), satisfying the industry standard of 10-years data retention. Such the excellent nonvolatile charge storage capability of InSe A-FET was mainly attributed to the boosted charge trapping and releasing process owing to the native oxidation of the InSe channel. The cycle endurance of the memory device was assessed by switching the Program and Erase

states between a low current level and high current level. As shown in Fig. 4e, the steady read currents as a function of cycle number illuminate its good reproducibility. Besides, Fig. 4f outlines the dynamic multilevel storage behaviour of InSe A-FET based memory by modulating the amplitudes of the erasing voltage pulses. Initially, the device was excited under a programming pulse of +80 V to store numerous electrons in the $InO_x$ trapping layer. Then the incremental read currents were clearly distinguished over almost five orders of magnitude under the application of 6 erasing voltage pulses, which resulted from the controllable amount of erased charges.

**Artificial synaptic features bsaed on InSe A-FET.** Inspired by the charge trapping dominated memory characteristics of InSe A-FET discussed above, we then investigated its basic synaptic functions for application as an artificial InSe synaptic device. For a fair comparison, the retention capacity evaluation and the synaptic behaviour emulation were also implemented on InSe V-FET device to further confirm the important role of the native oxidation effect for the efficient application of environmentally sensitive semiconductors (Supplementary Fig. 7). In Fig. 5a, the biological synapse serves as an important media to deliver information by sending chemical neurotransmitters from pre-neuron to post-neuron[59–64]. A suitable external stimulus from a pre-synaptic neuron would open the voltage-controlled $Ca^{2+}$ channel and lead to the released neurotransmitters passing through the synapse and arriving at the receptor on the post-synaptic neuron. Then, a sufficient number of neurotransmitters would trigger the opening of a chemical-controlled channel on the post-synaptic neuron and realize the signal transportation. Correspondingly, this phenomenon can be analogized as the gating-modulated carrier transport process in the InSe artificial synaptic device (the right side of Fig. 5a), in which the electrical input signal from the gate electrode is considered as the input spike in the pre-neuron to trigger the postsynaptic current (PSC), and the corresponding conductance in the InSe channel resembles the synaptic weight ($w$)[65]. Note that the generated PSC can be assigned as excitatory PSC (EPSC) or inhibitory PSC (IPSC), which is determined by the synaptic weight of the two connective neurons. To mimic the information transmission in a biological synapse between two neighbouring neuron cells, the EPSC responses of the InSe artificial synaptic device under several presynaptic $V_{bg}$ pulses with various amplitudes were first examined. As shown in Fig. 5b, all of the recorded EPSC curves for InSe A-FET underwent a rapid increase and a slow decay period when a presynaptic spike was applied. With the increase of the spike amplitude, they cannot return back to the initial current, signifying that the conductance of the InSe channel exhibited a memory feature at ambient conditions. The peak value of EPSC gradually increased along with the increase of the applied spike pulse from −40 to −80 V, which corresponded to the excitatory synaptic behaviour. While in the case of InSe V-FET, the sharp drop can be evidently observed from the measured current curves, offering self-consistent evidence for the worse retention characteristics of InSe V-FET (Supplementary Fig. 7a and b). The inset of Fig. 5b provides the calculated synaptic weight ($\Delta PSC/PSC$) depending on the spike amplitudes under two conditions. It is expected that the value of the voltage spike can be largely reduced to realize low energy consumption by matching a thinner dielectric layer or other dielectric materials. Figure 5c displays the variation of $w$ along with increasing the pulse width from 50 to 500 ms. Another dynamic evolution of the recorded IPSC responses is shown in Supplementary Fig. 8, which explained that more electrons were trapped in the $InO_x$ layer for a longer pulse duration.

The performance of learning and forgetting operations in a human brain relies on its capability to manage the synaptic weight, that is, the synaptic plasticity, which can be classified as STP and LTP according to the retention ability[66]. PPF is considered as a typical STP feature of a synapse, which is closely related to the synaptic activity for executing complicated neuronal tasks[67]. It describes the process when a neuron cell receives two consecutive actions, more neurotransmitters in synaptic vesicles will be released to result in the enhanced postsynaptic response. This phenomenon can be successfully implemented in our artificial synapse by simply applying a pair of presynaptic spikes and recording the corresponding variation of EPSC. The obtained PPF index depending on the time interval between two input spikes is shown in Fig. 5d. The value of the PPF index gradually decreased from 68 to 18% with the increasing time interval because of the enhanced releasing process of electrons from the $InO_x$ trapping layer, which could be well fitted by the double exponential function[68]. Note that the extracted relaxation time constants ($t_1 = 10$ ms, $t_2 = 320$ ms) coincide with the reported values in biological synapses, and the details can be found in Supplementary Fig. 9. In addition, Fig. 5e and Supplementary Fig. 10 exhibit the extracted IPSC as a function of retention time by varying the number of input spikes. If the input voltage is sufficiently large (+50 V), the recorded IPSC sharply decreased and then could not climb back to the initial state over the measurement periods even after undergoing few spike training, which illuminated the emulation of the LTP behaviour. Compared with that for InSe A-FET, the PPF index for InSe V-FET slightly changes near 0% with the increasing time interval (Supplementary Fig. 7c), instead of the exponential variation. Also, the recorded current curves under several sequential gate voltage pulses almost overlapped with each together, even under 50 sequential pulses (the inset of Fig. 5e), failing to demonstrate the LTP. These results obviously indicate that the synaptic features cannot be mimicked in the InSe V-FET system owing to the absence of the $InO_x$ interfacial layer to effectively boost the charge transport behaviour. Furthermore, sequential voltage pulses were executed as the presynaptic input signals to examine the dynamic potentiation and depression behaviour of the InSe artificial synaptic device. The monitored EPSC in Fig. 5f stepwise increased to $1.94 \times 10^{-5}$ A by varying the 50 pulse sequences of the negative voltage, and then dropped back to $7.9 \times 10^{-9}$ A by applying the positive voltage pulse sequences. The extracted synaptic weight changes depending on the sequential input pulses were provided in Supplementary Fig. 11. All of the positive and negative exponentially varied PSC responses showed the flexible plasticity of the InSe artificial synaptic device, corresponding to the sustained Erase and Program operations of electrons between the InSe channel and $InO_x$ trapping layer, respectively.

As one of the representative characteristics of LTP, STDP is of significance to the Hebbian synaptic learning and memory functions, in which the synaptic weight can be adjusted by the time interval and the spike order between the pre- and postsynaptic stimulations[13]. If the presynaptic spike arrives before that of the postsynaptic, that is $\Delta T > 0$, it will result in long-term potentiation. If $\Delta T < 0$, it will lead to long-term depression. In our work, the STDP behaviour of the InSe artificial synaptic device was built by applying the separated pre- and postsynaptic spikes from the back gate terminal and drain terminal, respectively. As shown in Fig. 5g, the changes of synaptic weight under $\Delta T > 0$ and $\Delta T < 0$ respectively indicated the potentiation and depression responses in our artificial synaptic device. The fitted curves based on the exponential functions located in the first and third quadrants were highly consistent with the excitatory feature in the typical biological synapse[69].

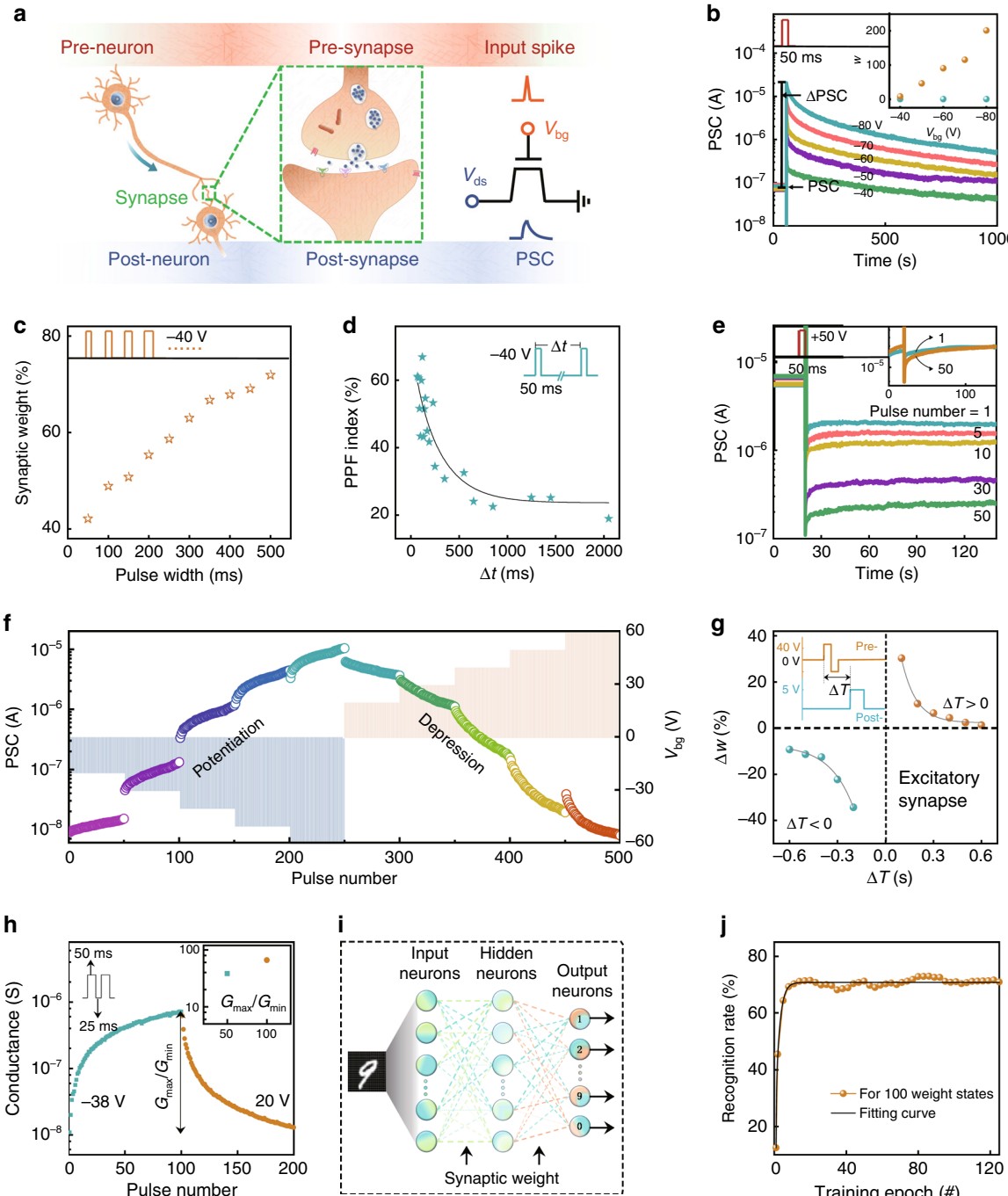

**Fig. 5 Emulation of the synaptic functions by the InSe artificial synaptic device. a** Schematic illustration of a biological synapse and the InSe artificial synaptic device. **b** EPSC generated by applying several input spikes with different voltage amplitudes under the potentiation condition of −80, −70, −60, −50, and −40 V, respectively. The drain voltage is set as 0.1 V under the Read operations. The change of PSC after the input spike (ΔPSC) is labelled by arrows. Inset shows the calculated $w$ for the case of InSe A- (Brown) and V-FET (Turquoise) devices. **c** Variation of $w$ under different pulse widths. **d** Extracted PPF index (($A_2 − A_1$)/$A_1$) versus spike time interval $\Delta t$, where $A_1$ and $A_2$ are the first and second EPSC peak, respectively. The solid line is the fitted curve based on the double exponential function. **e** The plot of IPSC changes over 120 s after stimulating by various numbers of the input pulse. The inset shows the corresponding changes after 1 and 50 pulses in InSe V-FET. **f** Monitored PSC under several sequential voltage pulses with various amplitudes (pulse width of 50 ms, time interval of 100 ms), which correspond to dynamic depression and potentiation behaviours in PSC. **g** Changes of synaptic weight (Δw) depending on the Time interval (ΔT) to emulate the STDP behaviour and the corresponding fitted curves based on the exponential functions. Note that the Δw is defined as ($\Delta PSC_2 − \Delta PSC_1$)/$\Delta PSC_1$, where $\Delta PSC_1$ and $\Delta PSC_2$ represent the obtained changes of PSC response after the pre- and postsynaptic simulations, respectively. The fitted lines are provided as a guide to the eye. The inset shows the schematic of the separated input signals from two terminals with the pulse width of 200 ms. **h** The potentiation and depression weight states of the conductance ($G$) extracted from InSe A-FET device under 100 successive input pulses. The inset shows the $G_{max}/G_{min}$ (conductance margin) under 50 pulses (38.1) to 100 pulses (65.3), respectively. **i** The schematic of ANN-based on InSe A-FET devices for image recognition. **j** The obtained recognition rates as a function of training phases for 100 weight states.

Furthermore, we constructed a learning platform based on a three-layer perceptron network model and the above InSe artificial synaptic devices to simulate the system-level pattern recognition[70]. As shown in Fig. 5i, the designed ANN consists of 784 input neurons, 150 hidden layer neurons, and 10 output neurons. The binarized images ($28 \times 28$ pixels) in the MNIST (Modified National Institute of Standards and Technology) handwritten database and the obtained digitals (0–9), serving as the image data vector and output vector, were assigned to the input and output layers, respectively. The variable weight connections among these neurons, as the synaptic weight vector, correspond to the potentiation and depression conduction states of InSe synaptic devices, which were experimentally extracted under successive input spikes (Fig. 5h). To perform the recognition tasks for digital images, 60000 images were used from the MNIST database during the learning process; the vector conversion via the sigmoid activation function and the weight update via the back-propagation learning algorithm were involved in. Note that the cycle-to-cycle weight update variation ($\sigma$) was set to 1%. Figure 5j shows the obtained recognition rates of the digitals (Supplementary Fig. 12 for 50 weight states) after 125 training epochs, with each epoch size of 8000 images. The overall accuracy rate improves from 45 to 70% with the increasing conductance margin, hinting at the positive effect of higher conductance margin on the recognition rate for MNIST patterns. On the other hand, compared with the previous works for the 2D materials-based electronic devices, the proposed InSe A-FET device presents three vital advantages: (1) The interfacial native charge trapping layer has little impact on electrical properties or stability; (2) The current on-off ratio and high mobility of InSe shows the potential for developing high-speed electronics; (3) The low power consumption systems can be expected through improving high-k dielectric materials or diminishing the channel length to deduce the spike duration time, spike voltage, as well as the response current. Thus we believe that the emulation of synaptic functions in device level and the simulation of pattern recognition in system-level in this work further demonstrate the important availability of native oxide-inspired van der Waals devices for designing promising neuromorphic architectures.

## Discussion

In summary, we developed a native oxide-inspired synaptic device based on the boosted charge trapping behaviours in a typical environmentally sensitive 2D material, InSe. Combining the HRTEM analysis and dynamic characteristic measurements showed that the thin layer of $InO_x$ located at the bottom of the InSe channel is responsible for the controllable charge trapping and releasing behaviour in InSe A-FET, which corresponds to surface dominated carrier fluctuations. Thus, the unique configuration is believed to provide innate opportunity to demonstrate reliable memory operations with multiple storage characteristics. Furthermore, synaptic functions at the device level based on the InSe artificial synaptic device by emulating the flexible plasticity of PPF and STDP at the device level, and the pattern recognition simulation at the system level based on the InSe artificial synaptic device, were successfully achieved, which pioneer a new opportunity for ultra-sensitive van der Waals electronics, such $MoTe_2$, $HfS_2$, or BP, to effectively control their native oxidation events towards establishing processing-efficient neuromorphic computing systems.

## Methods

**Fabrication of the InSe devices**. The few-layered InSe flake was mechanically exfoliated from the InSe crystal with the aid of polydimethylsiloxane (PDMS) films for better electrical performance, and then transferred onto the $SiO_2$/Si substrate. The thickness of the $SiO_2$ dielectric layer was selected to be 300 nm to benefit the analysis of the thickness of the InSe channel. Then, a 32-nm-thick In layer was deposited on the top of the InSe channel to achieve the excellent contact condition and high carrier mobility. Finally, a pair of 50-nm-thick Au electrodes were deposited by thermal evaporation at a pressure below $3 \times 10^{-6}$ Torr to form the source-drain contacts with the InSe channel. The AFM scan was implemented to examine the thickness of the exfoliated InSe flakes.

**Characterization of the electrical properties of the InSe devices**. The electrical performance of InSe A-FET and the emulation of synaptic functions were characterized under the ambient environment using a Lakeshore probe station equipped with a B1500A semiconductor parameter analyser. The electrical characteristics of InSe V-FET were also determined under vacuum environment for a better comparison. To examine the formation of the native oxidation $InO_x$ layer, the cross-sectional analysis of InSe A-FET was conducted using focused ion beam (FIB) systems and HRTEM technology, equipped with EDS. To explore the dominant mechanism of the carrier transport behaviour in the InSe V- and A-FET, the dynamic characteristic measurements were performed based on an in situ KPFM (Dimension Icon, Bruker) and a programmable point probe noise measurement system (3PNMS). For in situ Kelvin Probe Force Microscopy (KPFM) measurement (Bruker Dimension Icon SPM system), the conductive AFM probe (Pt/Ir) (AppNano, ANSCM-PA, 40 N/m, 300 kHz) was set as ground and the bias pulses (±20 V) were applied from the back-gate terminal via an external precision source unit (Keysight B2912A). The surface potential difference between tip and sample were detected under the tapping mode and the 2nd interleave scanning with the tip lift height 30 nm. The scanning range and resolution were 2 μm and 512 by 128 pixels respectively. The system noise floor is about $1 \times 10^{-27}$ $A^2$ $Hz^{-1}$. The source-drain current fluctuations were recorded at a certain gate voltage and source-drain voltage to analyze the dynamic carrier transport behaviour under ambient and vacuum conditions, respectively. Besides, for minimizing the external electrical interference to the monitored charge fluctuations, all the measurements were performed in a grounded metal cavity on an isolated table under the dark conditions.

## Date availability

The data that support the findings of this study are available from the corresponding author upon request.

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

## Acknowledgements

This work was financially supported by the National Key Research and Development Program of China (Grant No. 2016YFB0501604), the Natural Science Foundation of China (Grant Nos. 61774061 and 61504043), the NSAF Foundation of China (Grant No. U1830130), Natural Science Foundation of Shanghai (Grant No. 19ZR1473400), and the Taiwan Ministry of Science and Technology (Grant Nos. MOST 108-2112-M-005-012-MY3 and MOST-109-2636-M-009-002).

## Author contributions

F.-S.Y., M.J.L., S.-H.L. and Y.-F.L. designed the experiments and analyzed the data. F.-S.Y., M.-P.L. and I.-Y.H. contributed to the device fabrication. F.-S.Y. and M.J.L. performed the electrical characterizations. J.-Y.C., J.-K.C., S.-H.Y, Y.-M.C., K.-C.L. and Y.-C.C. contributed to the properties investigation of the materials. The InSe material was prepared by C.-H.H. H.F.L., Y.Z.L. and W.W.L. conducted the digit recognition simulation of the artificial neural network. M.J.L., F.S.Y. and Y.-F.L. prepared and revised the manuscript. All authors have discussed the results and commented on the manuscript.

## Competing interests

The authors declare no competing interests.
