## [Peer Review File · Nature Communications]

Reviewers' comments:

Reviewer #1 (Remarks to the Author):

The submitted paper reports long- and short-term memory effects of an InSe FET measured in air. The authors claim that the memory effect arises from an interfacial oxide layer (InOx) between the InSe channel and SiOx substrate. Subsequently, different memory effects depending on different programming methods are shown, which are compared with synaptic plasticity behaviors. Routine characterizations were performed as in a number of papers addressing very similar behaviors in different materials.

I do not see any convincing evidence for the role of the InOx interfacial layer in enlarging the conductance hysteresis. Do the authors assume that the interfacial layer was formed during the measurement in air? It should be clarified if the switching between different hysteretic behaviors depending on the atmosphere is reversible.

In fact, it makes little sense that the difference in Nit between V- and A-FET accounts for the difference in conductance hysteresis. First, the hysteresis direction is opposite; the V-FET has a clock-wise hysteresis while the A-FET has an anticlock-wise one. This explains that the number of Nit is not a direct cause of the difference in hysteresis. I suggest the authors to go through systematic device modeling to identify the effect of different Nit values on device hysteresis.

It is not clear in which sense the authors call the FET a neuron transistor. A neuron is a local processing unit that compares the current neuronal variable (mostly, membrane potential) with a given threshold and fires an event (spike) when the variable value exceeds the threshold.

Considering this essential role of a neuron in a spiking neural network, I do not understand the authors' terminology. If the authors want to show synaptic transmission using a set of their transistors, it is necessary to build a series of transistors and experimentally show the transmission results by varying the weight.

The synaptic features shown in Fig. 5 should be proven to arise from the charge trapping and detrapping in the interfacial oxide layer by rigorous device simulations.

Reviewer #2 (Remarks to the Author):

In this work done by F.-S. Yang et al., it is important to effectively utilize the inherent oxidation behavior in ultra-sensitive semiconductors to adjust their electronic properties and develop extended applications, especially in the case of device scaling. This manuscript presents an innovative design of native oxidation-inspired InSe transistor through a simple processing method and emulates the synaptic functions as an artificial synapse. The formation of the native oxide layer under ambient conditions is clearly examined by microscopic structure observation, and the boosted charge trapping and releasing process is confirmed by systematically dynamic characteristic measurements for the first time. This interesting study establishes a unique paradigm for environmentally sensitive van der Waals materials to build electric-modulated neuromorphic computing systems. Overall, I highly recommend it to be published in Nature Communications after minor revision:

1. The authors claimed that the native oxide layer of InSe is formed by HRTEM observation, which results in the distinct differences of transfer characteristics under vacuum and ambient conditions, as shown in Figure 1c and Figure 1d. If the InSe A-FET with a hysteresis loop in the transfer curve is stored back to the vacuum condition again, will the loop appear or not?
2. Can the authors provide more evidence, such as the transfer curves of InSe A-FET measured under vacuum conditions, to further prove the formation of oxide, instead of physical absorption?
3. As shown in Figure 4, the device doped by In layer demonstrates excellent nonvolatile memory performance, such as good Program/Erase current ratio and retention capacity. How about the device without the protection of In layer?
4. It is interesting that the basic electrical characteristics of InSe FETs can be largely improved

through surface doping method (Figure S1), whether the surface charge doping via In encapsulation is also effective for other layered materials?

5. Measurements of low-frequency noise are interesting, if it is possible, please offer the detail about measurement condition more.

The proposed device design and main conclusions elaborated in this manuscript are validly supported by the sufficient experimental results and comprehensive data analysis. The topic and the quality of the manuscript is well-matched with the scope of Nature Communications. I anticipate that the work can be published in this journal soon.

Response to Reviewer #1:

Reviewer #1: The submitted paper reports long- and short-term memory effects of an InSe FET measured in air. The authors claim that the memory effect arises from an interfacial oxide layer (InO_x) between the InSe channel and SiO_x substrate. Subsequently, different memory effects depending on different programming methods are shown, which are compared with synaptic plasticity behaviours. Routine characterizations were performed as in a number of papers addressing very similar behaviours in different materials.

We thank the Reviewer #1 sincerely for carefully reviewing our manuscript and we are very grateful to the Reviewer #1 for raising the professional suggestions and comments and for pointing out the misapprehensive points in our present manuscript in time. In this work, we successively conducted the basic hysteresis characterizations, the microscopic observation, and the dynamic characteristic measurements to deduce, to confirm, and to verify the native oxidation events and boosted charge trapping/detrapping behavior in InSe FET under different conditions, respectively; then inspired by such the unique interfacial layer, we performed the memory performance characterizations and the synaptic functions emulation. On the one hand, the adopted characterizations in our work, such as the basic **electrical measurements, HRTEM, especially the low-frequency noise** are recognized as the **most valid tools** in judging the in-depth physical mechanism of the charge transfer as well as the performance of electronics for practical applications and have been **widely used** in numerous outstanding works (Please refer to the works as *Nat. Nanotechnol.* **8**, 549 (2013); *Nat. Commun.* **4**, 1624 (2013); *Nat. Nanotechnol.* **12**, 901 (2017); *Nat. Mater.* **17**, 335 (2018); *Nat. Electron.* **1**, 130 (2019); *Nat. Commun.* **8**, 2121 (2017); *Nat. Electron.* **2**, 108 (2019).....). On the other hand, although a recent work reported that the oxide layer PO_x of black phosphorus FET would dominate its synaptic features (*Adv. Mater.* **28**, 4991 (2016)), **the underlying mechanisms of the charge transfer process have not been uncovered clearly yet**. Based on the Reviewer #1's valuable comments, more rigorous experiments (**1. The systematic comparisons of exposing-time dependent electric characteristics; 2. Atmosphere dependent hysteretic features; 3. The synaptic feature emulation based on InSe V-FET**) has been redesigned and conducted to clarify the role of InO_x interfacial layer in charge trapping/detrapping process and the significance of boosted charge trapping/detrapping events to synaptic features in InSe FET-based artificial devices. To make a clear explanation to the Reviewer #1, all proposed comments and suggestions have been carefully considered and answered point-by-point and the relative corrections are highlighted in the revised version of the manuscript.

Comment 1:

I do not see any convincing evidence for the role of the InO_x interfacial layer in enlarging the conductance hysteresis. Do the authors assume that the interfacial layer was formed during the measurement in air? It should be clarified if the switching between different hysteretic behaviors depending on the atmosphere is reversible.

Response 1:

The authors would like to thank Reviewer #1 to point out the professional and precious comments about the process of the formation of In oxide layer. In our present work, the InO_x layer, which results in the considerable hysteresis loop and boosted charge trapping and detrapping behaviour, was determined at the bottom of the InSe channel by TEM observation and dynamic characterizations. Based on the Reviewer #1's comments, we further designed the systematical experiments to rigorously study the reason of the formed oxide layer. The specific experiments were implemented as the following:

First, we measured the transfer curve of as-fabricated InSe FET under vacuum conditions. Second, we exposed the device into the ambient atmosphere for around 20, 40, 60, 80, and 100 minutes, respectively. Then, the same device was stored back to the vacuum conditions and recorded the exposing time (t_e)-dependent transfer curves after each exposing time. It is emphasized that all of the data records were implemented in vacuum to intentionally minimize the contribution of the oxide formation in air during the electrical measurement process. From the results shown in **Figure R1**, we can see that the hysteresis loop of transfer curves is enlarging along with the exposing time in air from $t_e= 20$ to 80 min, and then it is saturated at around 90 V for $t_e= 100$ min. This phenomenon indicated that the oxide interfacial layer can be formed when once exposing to air conditions, instead of during the measurement process in air. Besides, after immersing the device in the ambient atmosphere over 2 hours, a large hysteresis loop can also be well-retained when storing it back to vacuum conditions again. This result strongly suggests that the enhanced hysteretic feature is originated from the formation of InO_x layer in InSe A-FET, instead of the physical absorption, which is fully consistent with the results obtained by TEM analysis and dynamic measurements. Therefore, to avoid misunderstanding and make a clear description, the relative results about the exposing time-dependent transfer curves of InSe FET were added as **Figure S2** in the revised manuscript. The corresponding descriptions of "To preliminary verify the hypothesis of the native oxidation event, the specific experiments were intentionally designed to detect the variation of the transfer curves for InSe FET depending on the exposing time and immersing conditions. On the one hand, **Figure S2** shows the evolution of transport windows along with the exposing time under ambient conditions. Initially, a narrow window in the transfer curve for InSe FET can be observed under vacuum conditions. With the increase of the exposing time under ambient conditions, the obtained hysteresis loop gradually enlarged and then saturated with a window of about 90 V. Moreover, such a wide hysteresis loop under ambient conditions over 2 hours can still be well-retained without obvious variation, even when storing it back to vacuum conditions again, which eliminates the impact of the physical absorption of gas molecules." have been added in the revised manuscript on **Page 5**.

Figure R1. (a) Transfer characteristics for InSe FET at different exposing-time conditions. Note that all of the curves were measured under vacuum conditions after immersing in ambient conditions for various exposing times to minimize the occurrence of oxidation during the electrical measurement process in air. (b) Transfer curves recorded under ambient conditions (Brown) and vacuum conditions (Turquoise) after immersing in ambient conditions for 2 hours, respectively. Note that the drain-source voltage is set as 0.1 V.

Besides, the significant role of oxygen is further verified by investigating the electrical hysteretic features of InSe FET under various atmospheres, such as N_2 , Ar, and dry air. As we can see in **Figure R2**, both of the transfer curves measured under N_2 and Ar conditions deliver small hysteresis loops, which are fully-agreement with that of InSe V-FET. When immersing the same sample in the dry air atmosphere (30% O_2 and 70% N_2), the measured transfer curve shows an obvious hysteresis loop, which is the same as that of InSe A-FET, suggesting the relatively weak impact of moisture. Thus, it is believed that the boosted electrical hysteresis phenomenon in InSe A-FET can be mainly contributed to the formation of the native oxide layer under ambient conditions. Therefore, to provide more evidence and make a clear presentation, the measured transfer characteristics of InSe FET under various conditions have been added in **Figure S3**. The related descriptions “On the other hand, when storing a fresh sample in the nitrogen (N_2) or argon (Ar) atmosphere for around 30 minutes, the measured transport windows are almost fully-consistent with that of InSe V-FET (**Figure S3**). While in the case of dry air, it presents an enlarged hysteresis loop, even being stored back to the vacuum conditions, suggesting the weak influence of the hydrolysis effect on InSe channel. In consequence, it is believed that the enhanced hysteretic behaviour in InSe A-FET is dominated by the native oxide layer under ambient conditions.” have been added in the revised manuscript on **Page 5**. Thanks very much for the Reviewer #1 to raise these professional and constructive comments. The authors hope that our effort can offer an insightful understanding of the role of the oxide layer under ambient conditions in enhancing the hysteretic features of the InSe FET device to Reviewer #1.

Figure R2. (a) Transfer characteristics under N_2 for around 30 min and stored back to vacuum again. (b) Transfer characteristics under Ar for around 30 min and stored back to vacuum again. (c) Transfer characteristics under dry air (30% O_2 and 70% N_2) for around 30 min and stored back to vacuum again. The above results indicate that the enhanced hysteretic features are contributed from the native oxide layer, instead of the physical absorption. Note that the drain-source voltage is set as 0.1 V.

Comment 2:

In fact, it makes little sense that the difference in N_{it} between V- and A-FET accounts for the difference in conductance hysteresis. First, the hysteresis direction is opposite; the V-FET has a clock-wise hysteresis while the A-FET has an anticlock-wise one. This explains that the number of N_{it} is not a direct cause of the difference in hysteresis. I suggest the authors to go through systematic device modeling to identify the effect of different N_{it} values on device hysteresis.

Response 2:

The authors would like to thank for the Reviewer #1's comments about the sense of N_{it} difference and the hysteresis directions under different conditions. In our present work, along with the V_{bg} varying from -80 V to +80 V (forward sweep) and from +80 V to -80 V (reverse sweep), both of the directions in two transfer curves under vacuum and ambient conditions are clock-wise, which the same direction measured under ambient conditions wasn't marked in Figure 1d. Thus, to avoid misunderstanding and make a clear presentation, the corresponding arrows for the sweep directions under ambient conditions are shown in **Figure R3** and are added in the new version of the **Figure 1d** and **Figure 4c**.

Figure R3. (a) Transfer characteristics and the gate leakage curves at drain voltage $V_{ds} = 0.1$ V for the 15 representative InSe FETs under the ambient conditions. Note that the arrows mark the directions of the gate voltage sweep and ΔV_{th} means the threshold voltage shift. (b) Transfer characteristics of InSe A-FET with the V_{bg-max} (the maximum value of the swept range of gate voltage). The drain voltage is set as 0.1 V and the colourful arrows mark the increasing threshold voltage shift. Note that the arrows indicate the directions of hysteresis loops.

In addition, the InO_x layer formed under ambient conditions has been confirmed by TEM observation and dynamic characterizations, which is the core of the boosted charge trapping and detrapping behaviour in InSe A-FET. It is noticed that in our present back-gate device configuration, the extracted parameter of the effective trap densities, N_{it} , is composed of all the trap states from both the bottom interface and the top surface of the InSe channel. Please refer to our previous work, **Ref. 51** (*Adv. Mater.* **27**, 6612-6619 (2015)), to found more relative details of N_{it} in other 2D electronic devices. In our present work, it is evident that the difference of the origins of the trap states in ambient and vacuum conditions is attributed to the thin layer of InO_x . And the extracted value of N_{it} for InSe A-FET is around one order of magnitude higher than that for InSe V-FET. Thus, we believe that the average values of N_{it} under two conditions can be adopted to visualize the difference of the charge trapping and detrapping process in InSe A- and V-FET devices. Inspired by the Reviewer #1's comments, to make a clear description, the relative sentence of "Furthermore, the effective trap densities, N_{it} , were evaluated to visualize the process of charge transport in InSe FETs under the two conditions." has been revised as "Furthermore, the effective trap densities, N_{it} , which is composed of all the trap states from both the bottom interface and the top surface of the InSe channel in the present device configurations, were evaluated to visualize the process of charge transport in InSe FETs under the two conditions." and the sentence of "Such a big difference of N_{it} values between the two conditions sheds more light on the evident surface effect in InSe A-FET, resulting from the formation of 2-nm-thick native InO_x layer." has been revised as "Such a big difference of N_{it} values between the two conditions is believed to make the boosted charge trapping/detrapping events between the InO_x interfacial layer and InSe channel in InSe A-FET, shedding more light on the evident surface effect under ambient conditions because of the

formation of 2-nm-thick native InO_x layer.” in the new version of the manuscript on **Page 8**. Hope our explanation can provide a clear demonstration and offer Reviewer #1 an insightful understanding.

Comment 3:

It is not clear in which sense the authors call the FET a neuron transistor. A neuron is a local processing unit that compares the current neuronal variable (mostly, membrane potential) with a given threshold and fires an event (spike) when the variable value exceeds the threshold. Considering this essential role of a neuron in a spiking neural network, I do not understand the authors' terminology. If the authors want to show synaptic transmission using a set of their transistors, it is necessary to build a series of transistors and experimentally show the transmission results by varying the weight.

Response 3:

The authors would like to thank the Reviewer #1 for pointing out the comments about the terminology of the neuron-transistors. Generally, the biological synapse serves as the vital media to deliver information by sending chemical neurotransmitters from pre-neuron to post-neuron. For the development of neuromorphic computing, numerous efforts have been made in demonstrating the signal transmission from the nerve cell to the FET, or emulating the synaptic behaviors via “**solid-state electronics**”, during which generated the corresponding terminology, such as neuron transistors, neuro-transistors, or synaptic transistors. For instance, a neuro-transistor was developed by integrating dynamic pseudo-memcapacitors as the gates of transistors to generate electronic analogs of the soma and axon of a neuron (*Nat. Commun.* **9**, 3208 (2018)). A synaptic transistor, based on the ionic liquid-gated SmNiO₃ transistor on silicon platforms, has demonstrated non-volatile resistance and synaptic multilevel analogue states (*Nat. Commun.* **4**, 2676 (2013)). It states that a synaptic transistor is an electrical device that can learn in ways similar to a neural synapse. In addition, another recent work (*Small* **1**, 206 (2005)) about neuron transistor proposed that a prerequisite for the signal transmission from the nerve cell to the field-effect transistor (FET) is an adequate sensitivity of sufficiently small analog microelectronic devices in an aqueous environment. While in our present work, the gating-modulated carrier transport process in the as-prepared InSe A-FET based memory is analogized as an artificial synapse. What we focus on is whether the InSe based device with the unique advantages in the interfacial layer could mimic the basic synaptic functions, such as the short-term plasticity, paired-pulse facilitation, and long-term plasticity. As the schematic shown in Figure 5a in the main text, the electrical input signal from the gate electrode is considered as the input spike in pre-neuron, and the corresponding conductance in InSe channel is regarded as the synaptic weight, which can be effectively modulated by the gate voltage. Thus we called such the artificial synaptic device as a neuron transistor, which may be different from the understanding of the Reviewer #1. Therefore, considering the Reviewer #1's comments, in order to avoid misunderstanding, the name of “InSe neuron transistor” has been

revised as “InSe synaptic transistor” in the revised manuscript. Also to make a clear presentation, the title “Oxidation-boosted charge trapping in ultra-sensitive van der Waals materials for neuron transistors” has been revised to “Oxidation-boosted charge trapping in ultra-sensitive van der Waals materials for synaptic transistors”. The relative references have been added as Ref. 62 (Voelker, M. & Fromherz, P. Signal Transmission from individual mammalian nerve cell to field-effect transistor. *Small* **1**, 206-210 (2005)), Ref. 63 (Wang, Z. et al. Capacitive neural network with neuro-transistors. *Nat. Commun.* **9**, 3208 (2018)), and Ref. 64 (Shi, J., Ha, S. D., Zhou, Y., Schoofs, F. & Ramanathan, S. A correlated nickelate synaptic transistor. *Nat. Commun.* **4**, 2676 (2013)).

Besides, inspired by the Reviewer #1’s professional comments, the authors totally agree with the viewpoint that the synaptic transmission can be realized by designing the reasonable array of InSe transistors. It is of great significance for the development of neuroscience and neurocomputing, and we indeed make more efforts on it. However, on the one hand, the InSe FETs were fabricated by the typical method of mechanical exfoliation in our present work; on the other hands, few reports about the growth of large scale InSe thin film by other preparation method; thus it is still difficult to build the InSe FETs array to realize the signal transmission from nerve cell to FET. The authors would like to thank for the Reviewer #1’s constructive suggestions again about the methods of designing a set of InSe FETs for signal transmission, which offers a worthy issue for our subsequent studies. Hope that our explanation can offer a suitable understanding to the Reviewer #1.

Comment 4:

The synaptic features shown in Fig. 5 should be proven to arise from the charge trapping and detrapping in the interfacial oxide layer by rigorous device simulations.

Response 4:

The authors would like to thank the Reviewer #1’s constructive suggestions for that the synaptic features in InSe A-FET arisen from the boosted charge trapping/detrapping events should be further proven. In our present work, the enhancement of the hysteretic features in the transfer curves for InSe A-FET can be observed because of the formation of the native oxide of thin InO_x between InSe channel and SiO_2 . As confirmed by the HRTEM inspections, it can be believed that this amorphous interfacial layer can act as the effective trapping layer to boost the charge trapping/detrapping processes in the InSe A-FET-based memory system, which is highly consistent with the conclusion of low-frequency noise analysis. Thus, such a native oxide layer is considered as the necessity for InSe A-FET here to mimic the basic synaptic functions, such as short-term plasticity and long-term plasticity. Inspired by the Reviewer #1’s comments, to verify the vital role of the boosted charge trapping/detrapping events in the emulation of the synaptic functions, we further supplemented systematic experiments about the mimicking of the synaptic behaviour implemented on the InSe V-FET device. It is worth emphasizing that compared with the configuration of InSe A-FET, the native oxide of InO_x cannot be found in InSe V-FET, which has been demonstrated in our previous work (see *Adv. Mater.* **30**, 1803690 (2018)). The corresponding cross-section HRTEM image of InSe V-FET

in **Figure R4** is provided here to support our results. All of the operating conditions for the relative measurements shown in this supplementary experiment for InSe V-FET were intentionally as same as those for InSe A-FET for a fair comparison. The following are the specific experimental results:

Figure R5b shows the obtained postsynaptic current responses under several V_{bg} pulses with various amplitudes. As we can see, all of the recorded current curves underwent a rapid increase when applying a presynaptic spike, suggesting the excitatory synaptic behaviour. The corresponding current weights were extracted in the inset of **Figure R5c**. When taking the V_{bg} spike away, totally different from those in InSe A-FET, the current peaks drop quickly in each spike amplitude (-40 V, -60 V, and -80 V), which offer self-consistent evidence for the worse retention characteristics of InSe V-FET based memory in **Figure R5a**. Besides, we investigated the corresponding variation of postsynaptic current responses of InSe V-FET under a pair of input pulses with different time intervals (Δt). **Figure R5c** displays the calculated paired-pulse facilitation (PPF) index as a function of Δt . Compared with that of InSe A-FET, the values of PPF index for InSe V-FET exhibit the weak change near 0% along with the increasing Δt , instead of the exponential variation. This phenomenon significantly indicates that the synaptic features cannot be mimicked in the InSe V-FET system without the InO_x interfacial layer. Furthermore, **Figure R5d** shows the evolution of the current under the different numbers of input spikes. The recorded current curve under 50 sequential pulses almost overlaps with that under one pulse, demonstrating no signal of long-term plasticity. Thus, according to the systematical comparison of the experimental results between InSe V-FET and A-FET, we can conclude that the basic synaptic features mimicked by InSe A-FET in our present work are contributed to the native interfacial oxide layer, which dominates and boosts the charge trapping and detrapping process under gate bias.

Figure R4. (a) The Cross-section TEM image of the contact part for a typical InSe V-FET with 32-nm-thick In. The

scale bar is 10 nm. (d) High-resolution TEM images for InSe V-FETs with 8-nm-thick and 32-nm-thick In, respectively. And the EDS mapping for the contact part for the InSe V-FET with 32-nm-thick In. The scale bar is 5 nm. These results indicate that the oxide layer cannot be found in the case of InSe V-FET. (*Adv. Mater.* 30, 1803690 (2018))

Figure R5. (a) The retention ability of InSe V-FET based memory device. The readed currents were recorded at $V_{bg}=0$ V and $V_{ds}=0.1$ V. (b) The postsynaptic currents generated by the application of input spikes (-40 V to -80 V). Note that the drain voltage is set as 0.1 V under the Read operations. (c) The extracted values of PPF index depending on the increasing time interval, which show slight variations near 0 . (d) The generated postsynaptic currents under sequential simulation with various pulse numbers.

Figure R6. (a) Schematic illustration of a biological synapse and sketch of the InSe synaptic transistor. Note that the gate electrode serves as the presynaptic terminal and the current in the InSe channel is considered as the PSC response. (Courtesy of <https://tw.123rf.com> for creative commons attribution.) (b) EPSC generated by applying several input spikes with different amplitudes under the potentiation condition. The drain voltage is set as 0.1 V under the Read operations in this work. The change of PSC after the input spike (ΔPSC) is labeled by arrows. Inset shows the calculated values of w for the case of InSe A- (Brown) and V-FET (Turquoise) devices. (c) Variation of w under different pulse widths. (d) Extracted PPF index ($(A_2 - A_1)/A_1$) as a function of spike time interval Δt , where A_1 and A_2 are the first and second EPSC peak, respectively. The solid line is the fitted curve based on the double exponential function. (e) The plot of IPSC changes over 120 s after stimulation with a certain number of input spikes. For a better comparison, inset shows the corresponding changes after 1 and 50 pulses in the case of InSe V-FET. (f) Monitored PSC as a function of pulse number with varying amplitudes of sequential voltage pulses (pulse width of 50 ms, time interval of 100 ms). The application of the positive and negative voltage pulse sequences resulted in dynamic depression and potentiation behaviour in PSC. (g) Changes of synaptic weight (Δw) depending on the Time interval (ΔT) of the pre- and postsynaptic simulations and the corresponding fitted curves based on the exponential functions, which demonstrates STDP behaviour in the InSe synaptic transistor. Note that the Δw is defined as $(\Delta PSC_2 - \Delta PSC_1)/\Delta PSC_1$, where ΔPSC_1 and ΔPSC_2 represent the obtained changes of PSC response after

the pre- and postsynaptic simulations, respectively. The fitted lines are provided as a guide to the eye.

Therefore, to present more systematical results and make a clear presentation, **Figure 5** in our former version has been revised as **Figure R6** with more contrastive results and **Figure R5** has been added as **Figure S7** in the new version of the manuscript. The relative descriptions of “Inspired by the charge trapping dominated memory characteristics of InSe A-FET discussed above, we then investigated its basic synaptic functions for application as an InSe neuron transistor.” have been revised as “Inspired by the charge trapping dominated memory characteristics of InSe A-FET discussed above, we then investigated its basic synaptic functions for application as an InSe synaptic transistor. For a fair comparison, the retention capacity evaluation and the synaptic behaviour emulation were also implemented on InSe V-FET device to further confirm the important role of the native oxidation effect for the efficient application of environmentally-sensitive semiconductors (**Figure S7**).” The sentences of “As shown in Figure 5b, all of the recorded EPSC curves underwent a rapid increase and a slow decay period when a presynaptic spike was applied. The peak value of EPSC gradually increased along with the increase of the applied spike pulse from -40 V to -80 V, which corresponded to the excitatory synaptic behaviour. It is expected that the value of the voltage spike can be largely reduced to realize low energy consumption by matching a thinner dielectric layer or other dielectric materials. The inset of Figure 5b provides the calculated synaptic weight ($\Delta\text{PSC}/\text{PSC}$) depending on the spike amplitudes. Evidently, the EPSC was unable to return back to the initial current with the increase of the spike amplitude, which meant that the conductance of the InSe channel exhibited a memory feature.” have been revised as “As shown in Figure 5b, all of the recorded EPSC curves for InSe A-FET underwent a rapid increase and a slow decay period when a presynaptic spike was applied. With the increase of the spike amplitude, they cannot return back to the initial current, signifying that the conductance of the InSe channel exhibited a memory feature at ambient conditions. The peak value of EPSC gradually increased along with the increase of the applied spike pulse from -40 V to -80 V, which corresponded to the excitatory synaptic behaviour. While in the case of InSe V-FET, the sharp drop can be evidently observed from the measured current curves, offering self-consistent evidence for the worse retention characteristics of InSe V-FET (Figure S7a and b). The inset of Figure 5b provides the calculated synaptic weight ($\Delta\text{PSC}/\text{PSC}$) depending on the spike amplitudes under two conditions. It is expected that the value of the voltage spike can be largely reduced to realize low energy consumption by matching a thinner dielectric layer or other dielectric materials.” And the sentences of “Compared with that for InSe A-FET, the PPF index for InSe V-FET slightly changes near 0% with the increasing time interval (Figure S7c), instead of the exponential variation. Also, the recorded current curves under several sequential gate voltage pulses almost overlapped with each together, even under 50 sequential pulses (the inset of Figure 5e), failing to demonstrate the LTP. These results obviously indicate that the synaptic features cannot be mimicked in the InSe V-FET system owing to the absence of the InO_x interfacial layer to effectively boost the charge transport behaviour.” have been added in the new version of the manuscript on **Page 10-12**. Thanks very

much for the Reviewer #1's constructive comments and hope our effort and description can offer a clear explanation to the Reviewer #1.

Response to Reviewer #2:

Reviewer #2: In this work done by F.-S. Yang et al., it is important to effectively utilize the inherent oxidation behavior in ultra-sensitive semiconductors to adjust their electronic properties and develop extended applications, especially in the case of device scaling. This manuscript presents an innovative design of native oxidation-inspired InSe transistor through a simple processing method and emulates the synaptic functions as an artificial synapse. The formation of the native oxide layer under ambient conditions is clearly examined by microscopic structure observation, and the boosted charge trapping and releasing process is confirmed by systematically dynamic characteristic measurements for the first time. This interesting study establishes a unique paradigm for environmentally sensitive van der Waals materials to build electric-modulated neuromorphic computing systems. Overall, I highly recommend it to be published in Nature Communications after minor revision.

We thank the Reviewer #2 very much for carefully reviewing our manuscript, recognition of our work and for providing us with the professional and constructive comments that are of significance for further enhancing our work and improving the quality of the manuscript. According to the valuable suggestions, more rigorous experiments (**1. The systematic comparison of exposing-time dependent electric characteristics; 2. Atmosphere dependent hysteretic features; 3. The memory performance characterization for InSe FET without In layer deposition; 4. The verification of the enhanced surface doping effect on In₂Se₃ FET**) were performed and the manuscript was carefully revised. For convenience, each of the comments by the Reviewer #2 is answered point-by-point and all of the related corrections are highlighted in the revised version of the manuscript.

Comment 1:

The authors claimed that the native oxide layer of InSe is formed by HRTEM observation, which results in the distinct differences of transfer characteristics under vacuum and ambient conditions, as shown in Figure 1c and Figure 1d. If the InSe A-FET with a hysteresis loop in the transfer curve is stored back to the vacuum condition again, will the loop appear or not?

Response 1:

The authors would like to thank for the Reviewer #2 to point out the professional comment and totally agree with this viewpoint about examining the electrical performance of the InSe A-FET when storing it back to the vacuum conditions again. In our present work, the InO_x layer was determined at the bottom of the InSe channel by TEM observation and dynamic characterizations, which results in the considerable hysteresis loop and boosted charge trapping and detrapping behaviour under ambient conditions. For further verifying the formation of InO_x layer, it is of great significance to examine the electrical hysteretic features of InSe FET when it is stored back to the vacuum conditions again. Based on the Reviewer #2's comments, we designed a series of

experiments to rigorously study the reason for the formation of the oxide layer. First, we measured the transfer curve of as-fabricated InSe FET under vacuum conditions. Second, we exposed the device into the ambient atmosphere for around 20, 40, 60, 80, and 100 minutes, respectively. Then, the same device was stored back to the vacuum conditions and recorded the exposing time(t_e)-dependent transfer curves after each exposing process. It is emphasized that all of the data records were implemented in vacuum to in order to intentionally minimize the contribution of the oxide formation in air during the electrical measurement process. From the results in **Figure R7** we can see that the hysteresis loop in the transfer curves is enlarging along with the exposing time in the air increasing from $t_e= 20$ to 80 min, and then it is saturated at around 90 V for $t_e= 100$ min. This phenomenon indicated that the oxide interfacial layer can be formed when once exposing to air conditions, instead of during the measurement process in air. Besides, after immersing the device in the ambient atmosphere over 2 hours, a large hysteresis loop can also be well-retained when storing it back to vacuum conditions again. This result suggests that the enhanced hysteretic feature is originated from the formation of InO_x layer in InSe A-FET, instead of the physical absorption, which is fully consistent with the results obtained by TEM analysis and dynamic measurements. Therefore, to make a clear description, the relative results about the exposing time-dependent transfer curves of InSe FET were added as **Figure S2** in the revised manuscript. The corresponding descriptions of “To preliminary verify the hypothesis of the native oxidation event, the specific experiments were intentionally designed to detect the variation of the transfer curves for InSe FET depending on the exposing time and immersing conditions. On the one hand, **Figure S2** shows the evolution of transport windows along with the exposing time under ambient conditions. Initially, a narrow window in the transfer curve for InSe FET can be observed under vacuum conditions. With the increase of the exposing time under ambient conditions, the obtained hysteresis loop gradually enlarged and then saturated with a window of about 90 V. Moreover, such a wide hysteresis loop under ambient conditions over 2 hours can still be well-retained without obvious variation, even when storing it back to vacuum conditions again, which eliminates the impact of the physical absorption of gas molecules.” have been added in the revised manuscript on **Page 5**. Thanks very much for the reviewer to raising these professional and constructive comments. The authors hope that our effort can offer an insightful understanding of the role of the oxide layer under ambient conditions in enhancing the hysteretic features of InSe FET devices to the Reviewer #2.

Figure R7. (a) Transfer characteristics for InSe FET at different exposing-time conditions. Note that all of the curves were measured under vacuum conditions after immersing in ambient conditions for various times to minimize the occurrence of oxidation during the electrical measurement process in air. (b) Transfer curves recorded under ambient conditions (Brown) and vacuum conditions (Turquoise) after immersing in ambient conditions for 2 hours, respectively. Note that the drain-source voltage is set as 0.1 V.

Comment 2:

Can the authors provide more evidence, such as the transfer curves of InSe A-FET measured under vacuum conditions, to further prove the formation of oxide, instead of physical absorption?

Response 2:

The authors would like to thank for the Reviewer #2's constructive suggestions about the electrical properties of the InSe A-FET under vacuum conditions to support the formation of InO_x oxide layer, instead of physical absorption. In our present work, we use the TEM analysis and dynamic characteristic measurements to claim the existence of the oxidation layer and the boosted charge trapping and detrapping behaviour under ambient conditions. Based on the Reviewer #2's suggestions, more comparison experiments were conducted under various atmospheres, such as N_2 , Ar, and dry air, to further verify the formation of the native oxidation layer in InSe A-FET. The corresponding curves are shown in **Figure R8**. As we can see, both of the transfer curves measured under N_2 and A_2 conditions deliver small hysteresis loops, which are fully-agreement with that of InSe V-FET, as well as storing it back to the vacuum conditions. While immersing the same sample in the dry air (30% O_2 and 70% N_2), the measured transfer curve shows an obvious hysteresis loop, which is the same as that of InSe A-FET. Also, such a wide loop can still be retained consistently when measured under the vacuum conditions again. Thus, it is believed that the boosted electrical hysteresis phenomenon in InSe A-FET can be mainly contributed to the formation of the native oxide layer under ambient conditions, instead of physical absorption. Therefore, to provide more evidence and make a clear presentation, the measured transfer characteristics of InSe FET under various conditions have been added in **Figure S3**. Based on the reviewer's suggestions, the related

descriptions “On the other hand, when storing a fresh sample in the nitrogen (N_2) or argon (Ar) atmosphere for around 30 minutes, the measured transport windows are almost fully-consistent with that of InSe V-FET (**Figure S3**). While in the case of dry air, it presents an enlarged hysteresis loop, even being stored back to the vacuum conditions, suggesting the weak influence of the hydrolysis effect on InSe channel. In consequence, it is believed that the enhanced hysteretic behaviours in InSe A-FET are dominated by the native oxide layer under ambient conditions.” have been added in the revised manuscript on **Page 5**. Thanks sincerely for the reviewer to point out the constructive suggestions. Hope that our effort can offer a suitable explanation to the Reviewer #2.

Figure R8. (a) Transfer characteristics under N_2 for around 30 min and stored back to vacuum again. (b) Transfer characteristics under Ar for around 30 min and stored back to vacuum again. (c) Transfer characteristics under dry air (30% O_2 and 70% N_2) for around 30 min and stored back to vacuum again. The above results indicate that the enhanced hysteretic features are contributed from the native oxide layer, instead of the physical absorption. Note that the drain-source voltage is set as 0.1 V.

Comment 3:

As shown in Figure 4, the device doped by In layer demonstrates excellent nonvolatile memory performance, such as good Program/Erase current ratio and retention capacity. How about the device without the protection of In layer?

Response 3:

The authors would like to thank for the Reviewer #2’s valuable comments about the memory performance of the InSe FET without In doping layer (w/o In InSe FET). As previously reported, the electrical properties and the lifetime of InSe FET devices can be effectively enhanced by depositing an In doping layer on the top of InSe channel (Please refer to our previous work, *Adv. Mater.* **30**, 1803690 (2018)). Thus the excellent nonvolatile memory characterizations in our present work were recorded from the In layer doped InSe FET. Based on the Reviewer #2’s suggestions, to make a fair comparison and a clear explanation, the memory performance of w/o In InSe FET has been further investigated under ambient conditions and the corresponding results are shown as **Figure R9**. As we can see, different from that of In layer doped InSe FET under vacuum conditions, the transfer curve of w/o In

InSe FET initially exhibits a wider hysteresis loop because of the unavoidable oxidation during the fabrication process (Turquoise, Figure R9a). Figure R9b displays the charge retention capacity of w/o In InSe FET based memory. Here, the Program/Erase ratio drops from 10^3 to 3 over 2000 s, which indicates its worse charge retention ability compared with that of In layer doped InSe FET under ambient conditions. Besides, in Figure R9c, the continuous decay of reading currents as a function of cycle number suggests the terrible durability of w/o In InSe FET based memory. This phenomenon can be solidly explained that the severe oxidation of the surface of InSe channel would lead to the unstable electrical properties and poor charge retention ability under ambient conditions owing to the lack of effective protection from In doping layer, which matches well with the measured transfer curve after performing memory characterizations (Brown, Figure R9a). Hope our further effort and explanation could offer the Reviewer #2 a clear understanding.

Figure R9. (a) Transfer characteristics before and after memory characterizations of w/o In InSe FET under vacuum and ambient conditions, respectively. (b) The retention ability of w/o In InSe FET. The currents were separately read under Program and Erase states with $V_{ds}=0.1$ V and $V_{bg}=0$ V. (c) Endurance characteristics of the w/o In InSe FET memory device, which indicate the worse durability of the w/o In InSe FET based memory device. Note that the Program and Erase operations were carried out under cyclic voltage pulses and the Read state was operated under $V_{ds}=0.1$ V.

Comment 4:

It is interesting that the basic electrical characteristics of InSe FETs can be largely improved through surface doping method (Figure S1), whether the surface charge doping via In encapsulation is also effective for other layered materials?

Response 4:

The authors would like to thank the Reviewer #2 to point out the constructive comments about the availability of surface charge doping by In layer for other 2D materials. In our reported work, the In layer was deposited on InSe channel to act as the n-type doping layer and protection layer for InSe FET. The detailed surface doping mechanisms were systematically studied in **Ref. 43** (Please refer to *Adv. Mater.* **30**, 1803690 (2018)). Based on the Reviewer #2's suggestions, the controlled experiments about the In layer doped other layered materials, such as In_2Se_3 , have been conducted

under the same condition as InSe. In order to make a clear description, the related I - V characterizations have been provided in **Figure R10**. As we can see, after the deposition of about 30 nm_In layer, the transfer curves of layered In_2Se_3 FETs exhibit a high on-current level and remarkable n-type doping behaviour. Moreover, the extracted field-effect mobility of In-doped In_2Se_3 FETs reaches around $250 \text{ cm}^2 \text{ V}^{-1} \text{ s}^{-1}$, which is several times higher than that of normal In_2Se_3 FETs without In doping. Thus we could propose that the surface charge doping via In encapsulation also seems to be effective for layered In_2Se_3 FETs. Meanwhile, it should be emphasized that detailed analyses in subsequent works are certainly needed to fully comprehend the doping process of In layer to In_2Se_3 channel. Thanks sincerely for the Reviewer #2 to provide such significant comments. Hope that our effort can offer an insightful understanding of surface charge doping via In layer to the Reviewer #2.

Figure R10. (a) The transfer characteristics at room temperature of layered In_2Se_3 FET with the deposition of around 32-nm-thick In ($w/\text{In } \text{In}_2\text{Se}_3$ FET) as the doping layer at different V_{ds} , which indicates a strong n-type doping behaviour. The inset shows the optical micrograph images of the $w/\text{In } \text{In}_2\text{Se}_3$ FET. (b) The corresponding output characteristics of the same $w/\text{In } \text{In}_2\text{Se}_3$ FET at V_{bg} with the step size of 20 V.

Comment 5:

Measurements of low-frequency noise are interesting, if it is possible, please offer the detail about measurement condition more.

Response 5:

The authors would like to thank for the Reviewer #2's valuable suggestions. Generally, as to nanoscale devices, it is quite vital to understand the mechanism of the ratio of electric noise to device signal for practical applications in digital/analog electronics or sensors with a sensitive response, especially the signal fluctuations arising from the interface trapping layer or structural defects. In this work, we adopted the low-frequency noise technology, which is regarded as a

sensitive and useful tool to diagnose nanoscale electronics, to conduct dynamic characterizations for InSe A-FET to verify the boosted charge trapping behaviour derived from the formation of native InO_x layer. In particular, this characterization was carried out based on a programmable point probe noise measurement system (3PNMS). During the measurements, the source-drain current fluctuations were recorded at a given gate voltage and source-drain voltage to analyze the dynamic carrier transport behaviour. The system noise floor is about $1 \times 10^{-27} \text{ A}^2 \text{ Hz}^{-1}$. For minimizing the external electrical interference to the monitored charge fluctuations, all the dynamic characterization measurements were performed in a grounded metal cavity on an isolated table under the dark condition. Therefore, to make a clear presentation, the relative descriptions “To explore the dominant mechanisms of the carrier transport in the InSe V- and A-FET, the dynamic characteristic measurements were performed using a Programmable Point-Probe Noise Measuring System (3PNMS) under the dark condition.” have been revised as “To explore the dominant mechanism of the carrier transport behaviour in the InSe V- and A-FET, the dynamic characteristic measurements were performed based on a programmable point probe noise measurement system (3PNMS). The system noise floor is about $1 \times 10^{-27} \text{ A}^2 \text{ Hz}^{-1}$. The source-drain current fluctuations were recorded at a certain gate voltage and source-drain voltage to analyze the dynamic carrier transport behaviour under ambient and vacuum conditions, respectively. Besides, for minimizing the external electrical interference to the monitored charge fluctuations, all the measurements were performed in a grounded metal cavity on an isolated table under the dark conditions.” in the new version of the manuscript on **Page 15-16**. Hope that our effort can offer a suitable understanding to Reviewer #2.

Thank you in advance for your time and kind consideration of our manuscript.

Sincerely Yours,

Dr. Yen-Fu Lin,

Principal Investigator of Nano Research Group,

National Chung Hsing University (NCHU), Taiwan

2019 MOST Ta-You Wu Memorial Award Winner

Special commentator in *Nature Electronics*

Editorial Board Member in *Scientific Reports*

Editorial Board Member in *Physics Bimonthly*

Associate Professor, Department of Physics and Institute of Nanoscience, NCHU

Jointly Appointed Associate Professor, Research Center for Sustainable Energy and Nanotechnology,

NCHU

Website: <http://linyf.nchu.edu.tw/>

E-mail: yenfulin@nchu.edu.tw

Reviewers' comments:

Reviewer #1 (Remarks to the Author):

The authors have done additional experiments to justify the existence of the interfacial oxide layer which distinguishes the present transistor from the same authors' previous one. It feels that the interfacial layer is formed soon after its exposure to air (in a few tens of minutes). First, I am curious how the authors' previous transistor (published in *Advanced Materials*) kept it from oxidation completely. I assume that the transistor was supposed to be exposed to air at least a few tens of minutes unless all fabrication, characterization, and microstructure analysis were done in situ in vacuum. Please explain this.

It may be clear that the transistor includes the interfacial oxide layer; however, it is still not clear that the oxide layer causes the difference between the two transistors. That is, there may be a correlation between the interfacial layer and large hysteresis, which does not mean causality.

I am very uncomfortable with the authors' term "misunderstanding" in the response to my previous comment 3. The authors' transistor barely serves as a neuron in any sense. Neurons encode input spikes as output spikes following a particular coding scheme such as rate/temporal coding. Also, the basic principle of a neuron (or activation in deep learning) is integrate-and-fire, which is not clear in the proposed transistor. In any sense, the authors' transistor does not output spikes and encode input spikes as a form of spike trains.

If the authors believe the importance of the proposed transistor as a neuro-functional unit in comparison with previous works, I recommend the authors to validate the transistor such that network-level functionalities (prediction, classification, clustering) can be done with the proposed transistor in a better manner performance-wise (including energy efficiency) than previous devices.

Reviewer #2 (Remarks to the Author):

The authors have made significant modifications in the main text according to the reviewers' comments. All of the unclear points have been further confirmed by the supplemented experiments, especially for verifying the contribution of physical absorption to the formation of native oxide. Now the quality of this manuscript has reached the publication level. Therefore, I think this paper should be published soon in the current version without any further revision. As to the questions raised by Reviewer 1, in my opinion, an evident mistake had been made by Reviewer 1 about the opposite hysteresis directions under different conditions (Comment 2). I totally agree with the authors that the hysteresis directions are consistent under two different conditions and the values of the Nit are of great significance to evaluate the difference of the charge traps in two configurations.

In addition, although several works about the synaptic functions mimicked by layered devices have been reported recently, in this work, it further reported the underlying physical mechanism of the charge transport in a transistor via systematically dynamic characterizations, providing a new vista for the environmentally-sensitive electronics.

Reviewer #3 (Remarks to the Author):

The authors use the presence of native oxide on InSe to show synaptic behavior in InSe FETs. My major concern is in the synapse characterizations.

1. The authors show the variation of synaptic weight update with variable pulse width. The standard and algorithmically feasible approach is to use equal pulse width for all input pulses. How does the synaptic weight change when pulses of the same width are applied in succession?

2. In Fig. 5b, what are the various colors signifying? I assume those are for different pulse amplitudes? It needs to be clear. What are those amplitudes? Please put the label on the plot or in

the caption.

3. Same comment as above for Figure 5e.

4. What is the pulsing scheme for STDP of Fig. 5g? Please show.

Minor comment: Line 40-41: Should it not be molybdenum ditelluride?

Dear Reviewers,

We gratefully appreciate all comments from you and the Reviewers. Each reviewer's professional comments inspire our analytical thinking and significant modification of the present version of the manuscript. Based on the Reviewers' comments, we have further done the **key experiments and simulation**, such as the overall extracted charge trap densities in both devices and the in-situ KPFM measurement for examining the dynamic charge trapping event in **Figure R1**, the simulation of an artificial neural network for pattern recognition based on the InSe FET device in **Figure R2**, the evolution of the synaptic weights under successive input pulses in **Figure R3**, the supplementation of the experimental details for mimicking the artificial synaptic features in **Figure R5** and **Figure R6**, and survey relative works, such as the theoretical analysis of the physisorption binding energies of O₂, or the amorphous oxides induced trap states, to support our results. Then we revised our manuscript carefully in an all-round way and the corresponding supplementary. A list of changes in our revision and our reply is given as follows.

List of changes: (All of these changes are highlighted in the main text.)

1. **The original title of this work** has been changed to be "Oxidation-boost charge trapping in ultra-sensitive van der Waals materials for artificial synaptic features" **to give a more appropriate target about our current researches in solid-state electronics.**
2. **All of the segments** "synaptic transistor" in the main text have been revised as "synaptic features in device level" or "artificial synaptic device" **to make a suitable presentation.**
3. **Abstract: add the segment** "The pattern recognition capability of the InSe FET based artificial neural network".
4. **Paragraph 1, Page 2:** the segment "tellurium molybdenum" has been revised to "molybdenum ditelluride".
5. **Paragraph 2, Page 3:** the sentences "In this work" have been revised to "In this work, we.....for limiting its negative impact even utilizing it for enhancing the performance and application of electronic devices." **to further emphasize the importance of developing reliable methods to analyze and understand the charge transport process in native oxide layer for enhancing the performance and application of electronics.**
6. **Paragraph 1, Page 4:** add the segment "as well as the system-level pattern recognition based on the artificial neural network."
7. **Paragraph 2, Page 4:** the sentence "Such a slight shift" has been revised to "Such a slight positive shift of the threshold voltagesor fabrication-induced defects." **to determine the dominant effects for hysteresis loops.**
8. **Paragraph 2, Page 8:** the sentences "The obtained N_{it}in the vacuum condition." have been revised to "For a better comparison, the extracted values....., at least an order of magnitude at least higher than that in the vacuum condition."

9. **Paragraph 2, Page 9:** the sentences “On the other hand, we further conducted the in-situ.....and artificial synaptic features.” **to verify the dominant role in charge trapping effect of InO_x by in-situ KPFM measurements.**
10. **Paragraph 1, Page 13:** insert a sentence “The extracted synaptic weight changes depending on the sequential input pulses were provided in Figure S11.”
11. **Paragraph 2, Page 13:** The sentences “Furthermore, the fitted curves” have been revised as “The fitted curves based onin the typical biological synapse.”
12. **Paragraph 1, Page 14:** add a new paragraph “Furthermore, we constructed a learning platform based on a three-layer perceptron network model.....for designing promising neuromorphic architectures” to supplement **the system-level simulation results for image recognition** and conclude the **significance of this work and the advantages of InSe based systems compared with other similar 2D materials.**
13. **Paragraph 1, Page 16:** the sentence “Furthermore, synaptic functions.....” has been revised as “Furthermore, synaptic function at the device level.....computing systems.”
14. **Paragraph 2, Page 17:** insert new sentence of the KPFM measurement “For in-situ Kelvin Probe Force Microscopy (KPFM) measurement.....and 512 by 128 pixels respectively.”
15. **References [49]-[53] and [75]** have been added to demonstrate **the significant impact of the oxide layer dominated charge trapping/detrapping behavior on the performance and the potential applications of electronics,** and the information of pattern recognition simulation.
16. **Figure 3:** update the **values of N_{it}** and add more information about **the KPFM measurements to verify the charge trapping behaviour in the InO_x layer.**
17. **Figure 5:** add more information about **the experimental details during the synaptic characterizations in it to make a better presentation of the obtained results, and the new results of the system-level simulation results based on InSe FET devices for pattern recognition.**
18. **Add Figure S11** into Supporting information **for examining the evolution of the synaptic weight under successive input pulses.**
19. **Add Figure S12** into Supporting information to supplement the **simulation results of the recognition rate of the designed ANN system for 50 artificial synaptic weight states.**

Response to Reviewer #1:

Reviewer #1: The submitted paper reports long- and short-term memory effects of an InSe FET measured in air. The authors claim that the memory effect arises from an interfacial oxide layer (InO_x) between the InSe channel and SiO_x substrate. Subsequently, different memory effects depending on different programming methods are shown, which are compared with synaptic plasticity behaviours. Routine characterizations were performed as in a number of papers addressing very similar behaviours in different materials.

We sincerely apologize to the respected Reviewer #1 for our unapt expression and we also deeply thank sincerely for Reviewer #1's professional suggestions and comments about the terminology in the neuromorphic science, which are much valuable for us to precisely express our work and improve the quality of the manuscript. Compared to the recently reported works about the similar native oxidation phenomena (for example black phosphorus or other layered materials), where the mechanisms for charge transport is not clearly discussed, in this work, we carefully elucidated the physical mechanisms of intrinsic charge trapping/detrapping manners derived from the native oxide layer in the InSe FET. More importantly, despite an intuition in poor device operation due to the existence of a native oxide layer in the layered channel, we further demonstrated its potential applications in nonvolatile-memory and artificial synaptic functions for the first time. We believe that to quantitatively analyze and comprehend electron traps in the native oxide layer is of great significance for van der Waals materials to further limit its negative impact, even utilize it for enhancing the performance and applications of electronic devices. Based on Reviewer #1's valuable comments, more rigorous thinking and modifications (including **1. The title of this work; 2. The descriptions about the artificial synaptic devices; 3. The part of the Introduction; 4, The overall extracted charge trap density; 5, The charge trapping observation by in-situ KPFM; 6. The simulation results on system level for pattern recognition; 4. The advantages of the InSe based systems**) have been made to clarify the important role of InO_x interfacial layer in charge trapping/detrapping events and the potential in emulating artificial synaptic features of InSe FET-based devices. To make a clearer explanation to Reviewer #1, all of the comments have been carefully considered and answered point-by-point and the relative corrections are highlighted in the revised version of the manuscript.

Comment 1:

The authors have done addition experiments to justify the existence of the interfacial oxide layer which distinguishes the present transistor from the same authors' previous one. It feels that the interfacial layer is formed soon after its exposure to air (in a few tens of minutes). First, I am curious how the authors' previous transistor (published in *Advanced Materials*) kept it from oxidation completely. I assume that the transistor was supposed to be exposure to air at least a few tens of mins unless all fabrication, characterization, and microstructure analysis were done in situ in vacuum. Please explain this.

Response 1:

The authors would like to thank Reviewer #1's professional comments about the oxidation phenomenon of the InSe FET in our previously reported work in *Advanced Materials*. First of all, we totally agree with Reviewer #1's questions about the unavoidable contact between the InSe channel and the air (O_2 or others) during the fabricated processes. This is a good and important point: **the differences in exposure times and exposure modes in the air of the devices between two works**. In detail, our previous work focused on the evident n-type doping effect on InSe FET by the deposition of the indium (In) layer. In the fabricated process, we transferred fresh InSe flakes onto a substrate once exfoliated (in 5 minutes at most) and then they were almost protected in the vacuum conditions in the following steps, such as In layer deposition, electrode metallization, as well as electrical measurements. That means, each inevitable expose time gap between two adjacent stages, such as between the metal metallization and measurement, was almost controlled below 5 minutes and then the devices were stored back to the vacuum again. Here, we totally believe that as Reviewer #1's comment, in each short exposure time, the physisorption of O_2 on InSe channel cannot be avoided. While more reacts between them almost be broken once be stored back to the vacuum conditions due to the low physisorption binding energies (Please refer to the related paper for similar materials: Exploring the air stability of $PdSe_2$ via electrical transport measurements and defect calculations. *npj 2D Mater. Appl.* 2019, **3**, 50), which leads to the small hysteresis current loop. However, in our present work, the fabricated InSe FET was continuously exposed in the ambient conditions (30 minutes at least) for aging, which is believed to result in a dissociative adsorption/reaction of the O_2 molecules to atomic O and subsequent InO_x formation, as evidenced by the HRTEM observation and the enlarged hysteresis current loops along with aging time in ambient conditions. Similar oxidation phenomena were observed in other environmentally sensitive materials, such as black phosphorus. A relative work (**Ref. 14: *Adv. Mater.* 2016, **28**, 4991**) reported that a 2-nm PO_x layer on the bottom side of the BP flake can be formed in ambient conditions in **30 minutes**, which highly supported our observed results.

Secondly, it is worth emphasizing that in our previous work, the tested devices also exhibited the increasing hysteretic behavior along with the aging time, even being stored in a simple oxygen-proof drying box, which exactly motivated us to explore the latent native oxidation effect in present work. Furthermore, in the former response letter, the evolution of the transfer characteristics for InSe A-FET depending on the various exposing-time was carefully examined. The detailed experiment was conducted as: at first, a newly fabricated device was aged in air conditions for 20 minutes; then the corresponding electrical behavior was examined under vacuum conditions to minimize possible oxidation effect during the measurement process; after the measurement in a vacuum, it was rapidly taken into the ambient condition for proceeding the following aging. During such continuous circulations between vacuum measurement and ambient air exposure, more O_2 would be absorbed suddenly at the moment after measurement and entering into the ambient environment. Thus, the oxidation events would be potentially promoted than that stored in the ambient conditions all the

time.

Therefore, **combining both the experimental results** (in our previous and present works as well as the similar formation of PO_x for layered BP flakes) and **theoretical analyses** (the calculated physisorption binding energies), the native oxidation can be ensured in our InSe FET, which is the key functional layer for enhancing the charge trapping/detrapping process. Based on the Reviewer #1's comment, for making a clear presentation, the relative descriptions "In this work, we experimentally demonstrate an oxidation-boosted synaptic transistor, for the first time, based on a typical sensitive van der Waals semiconductor, InSe. As a representative of the III-VI group materials, InSe possesses small effective electron mass and excellent intrinsic charge transport characteristics. Its acknowledged air-instability and large surface to volume ratio, resulting in severe performance degradation, provide itself favorable conditions for introducing into oxidation layer" have been revised as "In this work, we experimentally demonstrate an oxidation-boosted artificial synaptic device, for the first time, based on a typical sensitive van der Waals semiconductor, InSe. As a representative of the III-VI group materials, InSe possesses small effective electron mass and excellent intrinsic charge transport characteristics. Its acknowledged air-instability and large surface to volume ratio, resulting in severe performance degradation and hysteretic behavior, attract much researchers' attention on how to protect it from contact with air for steady electrical properties, including our team. To put this in perspective, the environmental sensitivity provides itself favorable conditions for introducing into oxidation layer; thus quantitatively analyzing and comprehending carrier traps in this thin oxide layer is of great significance in particular for van der Waals materials to further limit its negative impact, even utilize it for enhancing the performance and applications of electronic devices." in the revised manuscript on **Page 3**. The relative references have been added as **Ref. 53** (Exploring the air stability of PdSe_2 via electrical transport measurements and defect calculations. *npj 2D Mater. Appl.* 2019, **3**, 50).

We sincerely hope our effect and explanation can offer a more suitable explanation to Reviewer #1. Besides, we would like to thank Reviewer#1 again for pointing out this valuable comment, which provides more struggling directions for our future works.

Comment 2:

It may be clear that the transistor includes the interfacial oxide layer; however, it is still not clear that the oxide layer causes the difference between the two transistors. That is, there may be a correlation between the interfacial layer and large hysteresis, which does not mean causality.

Response 2:

The authors would like to thank for the Reviewer #1's comments about the causality between the interfacial oxide layer and the current hysteresis. In general, the hysteretic behaviors in transfer characteristics of a low-dimensional system are originated from **two main effects**: one is the charge transfer between the conducting channel and interfacial layer; another is the opposite charges in

the channel derived from the charges in the interfacial layer (Please refer to Artificial optic-neural synapse for colored and color-mixed pattern recognition. *Nat. Commun.* 2018, **9**, 5106.; A Dynamically reconfigurable ambipolar black phosphorus memory device. *ACS Nano* 2016, **10**, 10428.; Hysteresis of electronic transport in graphene transistors. *ACS Nano* 2010, **4**, 7221.; Graphene dynamic synapse with modulatable plasticity. *Nano Lett.* 2015, **15**, 8013.). In the former condition, the charge transfer leads to the positive shift of the V_{th} in backward gate sweeping because of the trapped charges by the interfacial layer. While in the latter one, the capacitive gating would result in a negative shift due to the enhanced local electrical field by dipoles in the dielectric layer. For example, an outstanding work by Prof. Park and Prof. Wong (*Nat. Commun.* 2018, **9**, 5106) proposed that the hysteresis characteristic in the h-BN/WSe₂ synaptic device was originated from the charge trapping layer between h-BN and WSe₂. It stated that “The hysteresis characteristic was observed in the current between the presynaptic and postsynaptic terminals, which was dependent on the voltage applied to the synaptic cleft terminal (V_{SCT}). This occurs because charges trapped in the WCL partially screen V_{SCT} and thereby influence the current flow through the synaptic device”, which supports our conclusions. Thus, based on the transfer curves of InSe A-FET in this work, the V_{th} in the backward sweeping presented an evident positive shift, which can be ascribed to the charge transfer effect.

In our present work, as proposed by Reviewer #1, an amorphous oxide layer (InO_x) was formed in the InSe A-FET under ambient. Combining the abovementioned charge transfer effect, the formed amorphous oxide layer and the obtained electrical properties for both InSe V- and A-FET, we preliminarily deduced that the evident difference in current hysteresis loops was contributed to the charge transfer effect between the interfacial oxide layer and the channel. And we also carefully adopted a series of methods, such as HRTEM and low-frequency noise technologies to examine the location of the oxide layer and the enhanced charge trapping/detrapping behavior in InSe A-FET. Furthermore, for verifying the charge trapping effect of InO_x, we captured the dynamic variation of the surface potential in both devices by in-situ Kelvin probe force microscope (KPFM) measurements. The program and erase process was operated via applying a +20 V and a -20 V gate voltage pulse (5 s), respectively. As the results in **Figure R1f**, the recorded potential of InSe channel in InSe A-FET exhibits the distinct difference between two read states (gate voltage= 0 V) after the successive program and erase operations and steady repeatability in 4 cycles, shedding light on the occurrence of the charge trapping/detrapping events. In contrast, the recorded potential in InSe V-FET was almost constant under several read states, indicating the weak charge transfer phenomena. These dynamic behaviours visually indicate the dominant role of the InO_x layer in trapping/detrapping electrons in the proposed devices, which paves the way for the demonstration of the artificial synaptic features. It is worth mentioning that, we totally agree with the Reviewer#1's comment that the causes for the charge transfer behavior also contains the contributions from SiO₂ interface or the unavoidable absorption on the top surface of InSe channel. While based on the evident differences between the InSe V- and A-FET, including device configurations, measurement

conditions, electrical characteristics, and surface potential variations, the significant role of the interfacial oxide layer for leading to the boosted charge transfer behavior can be determined. Also, the similar phenomena about the oxide layer induced enhancement of charge trapping behaviour had been demonstrated in the BP FET devices (Ref. 14: *Adv. Mater.* 2016, **28**, 4991), in which it has to be emphasized that the related mechanisms about trapping events were not systematically investigated.

Figure R1. **a**, Recorded S_1 as a function of frequency and gate voltages for InSe A-FET(upper). Note that a S_1 curve at $V_{bg}=60$ V based on the white dash line (upper) is profiled a representative of $1/f$ noise (under). **b**, Drain current normalized S_1 depending on I_{ds} for InSe A-FET at several V_{ds} . **c**, Box-plots of the fitted α (upper) and β (lower) values for InSe V- and A-FET. **d**, Normalized S_1 by I_{ds}^2 (discrete dots) versus I_{ds} on a log-log scale for InSe FET under different conditions. The dashed lines are the well-fitted results based on the correlated mobility fluctuation. **e**, The extracted values of effective interface trap density for InSe V- and A-FET as a function of z/λ and gate bias. **f**, The dynamic variations of the surface potential (under read states) of InSe channel by in-situ KPFM depending on the scanning time, for InSe A- and V-FET, respectively. The program and erase operations were defined by applying a +20 V and a -20 V gate voltage pulse (5 s) for 4 cycles.

On the other hand, based on the reported researches (Please refer to the relative work: Intrinsic charge trapping in amorphous oxide films: status and challenges. *J Phys.: Condens. Matter* 2018, **30**, 233001), the amorphous metal oxide layers, such as HfO_2 , Al_2O_3 , TiO_2 , contain the intrinsic deep electron and hole trap states stemming from the disordered network by experimental and

theoretical evidence. Therefore, developing reliable methods for analyzing and understanding the electron traps in thin oxide layers is believed of great significance for limiting their negative impact even utilizing them to enhance the performance and applications of electronic devices, such as memories, high sub-threshold slope transistors. For making a clear demonstration, as shown in **Figure R1e**, we extracted the values of N_{it} depending on gate bias in InSe A-FET and V-FET to emphasize the improved charge trap states and enhanced charge trapping capacity in InSe A-FET device.

Therefore, based on the Reviewer#1's professional comments, to make a clear description, the sentence "Such a slight shift of the threshold voltages below 20 V was attributed to the few interface states between the channel and dielectric layer or fabrication-induced defects." has been revised as "Such a slight positive shift of the threshold voltages below 20 V in backward gate sweeping can be attributed to the charge transfer effect instead of capacitive gating effect due to the few interface states between the channel and dielectric layer or fabrication-induced defects." in the revised version on **Page 4**. The related description of KPFM measurements has been supplemented as "On the other hand, we further conducted the in-situ KPFM measurements to visually verify the charge trapping effect of InO_x via examining the dynamic variation of the surface potential of InSe channel in InSe A- and V-FET (Figure 3f). In the case of InSe A-FET, the recorded potential of InSe channel exhibits distinct difference under the read states ($V_{bg}= 0$ V) after the program and erase pulses, as well as a good reproducibility in 4 successive cycles, which shed light on the occurrence of the charge trapping/detrapping events in InSe A-FET during the program/erase process. In contrast, the potential difference in InSe V-FET could almost be ignored under several read states, hinting at its weak charge transfer behaviour. Thus we could determine the dominant role of the InO_x layer in trapping/detrapping electrons in the proposed devices, which paves the way for the demonstration of memory and artificial synaptic features." in the new version on **Page 9**. The results of LFN and KPFM analysis (**Figure R1**) were updated in **Figure 3**. The KPFM measurement parameters have been added in the **Methods** parts as "For in-situ Kelvin Probe Force Microscopy (KPFM) measurement (Bruker Dimension Icon SPM system), the conductive AFM probe (Pt/Ir) (AppNano, ANSCM-PA, 40N/m, 300 kHz) was set as ground and the bias pulses (± 20 V) were applied from the back-gate terminal via an external precision source unit (Keysight B2912A). The surface potential difference between tip and sample were detected under the tapping mode and the 2nd interleave scanning with the tip lift height 30nm. The scanning range and resolution were 2 μm and 512 by 128 pixels respectively." The relative references have been added as **Ref. 49** (Artificial optic-neural synapse for colored and color-mixed pattern recognition. *Nat. Commun.* 2018, **9**, 5106), **Ref. 50** (Hysteresis of electronic transport in graphene transistors. *ACS Nano* 2010, **4**, 7221), **Ref. 51** (A Dynamically reconfigurable ambipolar black phosphorus memory device. *ACS Nano* 2016, **10**, 10428.), and **Ref. 52** (Intrinsic charge trapping in amorphous oxide films: status and challenges. *J Phys.: Condens. Matter* 2018, **30**, 233001) to support our results. We hope our explanation and effort can offer a more suitable explanation to Reviewer #1.

Comment 3:

I am very uncomfortable with the authors' term "misunderstanding" in the response to my previous comment 3. The authors transistor barely serves as a neuron in any sense. Neurons encode input spikes as output spikes following a particular coding scheme such as rate/temporal coding. Also, the basic principle of a neuron (or activation in deep learning) is integrate-and-fire, which is not clear in the proposed transistor. In any sense, the authors' transistor does not output spikes and encode input spikes as a form of spike trains.

If the authors believe the importance of the proposed transistor as a neuro-functional unit in comparison with previous works, I recommend the authors to validate the transistor such that network-level functionalities (prediction, classification, clustering) can be done with the proposed transistor in a better manner performance-wise (including energy efficiency) than previous devices.

Response 3:

The authors would like to thank Reviewer #1 for pointing out this query in time about the terminology of the neuron-transistors and totally agree with this viewpoint. We sincerely apologize to the respected Reviewer #1 for our unapt expression and the use of the neuron-transistor/the artificial synaptic device. For us, this is a very valuable comment to precisely present our work and improve the quality of the manuscript. In neuromorphic architectures, the biological synapse is regarded as the vital media for information transformation through chemical neurotransmitters released from the pre-synapse to the post-synapse. The words of "In principle, modern semiconductor technology could provide suitable tools for massive parallel monitoring of neuronal activity at high spatial and temporal resolution." have been proposed by recent studies (Ref. 66: *Small* 2005, 1, 206-210). It states that a prerequisite for the signal transmission from the nerve cell to the field-effect transistor (FET) is an adequate sensitivity of sufficiently small analog microelectronic devices in an aqueous environment. It means that the firing neuron is directly attached to the exposed gate oxide of a FET, in which the change of the local extracellular potential in the electrolyte of the cleft between cell and chip directly modulates the source-drain current. Recently, various solid-state electronics are employed to emulate synapse-like behaviour by demonstrating the controllable current weight via electric or light signals for the development of neuromorphic computing. In our present work, we mimicked the basic functions of an artificial synapse in device level by InSe A-FET, such as the PPF, STP, LTP, and STDP. The authors sincerely thank Reviewer #1's valuable suggestions here and we are deeply aware of the mistake in terminology and strictly corrected the name of "InSe neuron transistor" to "InSe A-FET based artificial synapse in device level" and all of the corresponding descriptions in the revised manuscript. For making a clearer description, the title "Oxidation-boosted charge trapping in ultra-sensitive van der Waals materials for neuron transistors" has been revised to "Oxidation-boosted charge trapping in ultra-sensitive van der Waals materials for artificial synaptic features". Here, we carefully apologize again to our respected Reviewer #1 for our unapt description in the former response letter. We totally understand that our clumsy use in language presentation has to be strictly forbidden and

be condemned. Thank a million for the respected Reviewer #1.

Motivated by Reviewer #1's professional comments, we totally agree with the viewpoint that the demonstration of the neural network-level functionalities, which is one of the research trends for neuromorphic engineering, is of great importance based on our InSe FET devices. As discussed in this work, our flakes made of a few layers of InSe were obtained by mechanical exfoliation of semiconducting InSe bulk crystal. The weak point of the exfoliation method is the small and uncontrollable lateral size of samples, but intrinsically physical behaviour can be directly addressed. In the current technology, how to grow a CVD large-scale InSe thin film is still a challenge, resulting in the limit of related reports. In this regard, we constructed a learning platform based on a perceptron network model and our InSe artificial synaptic devices to simulate the system-level pattern recognition. The MNIST (Modified National Institute of Standards and Technology) database was employed to perform the recognition tasks for digital images. As shown in **Figure R2b**, the designed artificial neural network (ANN) based on a three-layer perceptron model consists of 784 input neurons, 150 hidden layer neurons, and 10 output neurons. The binarized images (28×28 pixels) and the obtained digitals (0-9) were assigned to 400 input neurons and 10 output neurons, respectively. The variable weight connections among these neurons correspond to the potentiation and depression conduction states of InSe synaptic devices, which were experimentally extracted under successive input spikes (**Figure R2a**). Note that the G_{\max}/G_{\min} (conductance margin) increases from 38.1 to 65.3 at 50 and 100 pulses, respectively. During the learning process, 60000 images were used from the MNIST database; the processes involved in the vector conversion via the sigmoid activation function and the weight update via the back-propagation learning algorithm. **Figure R2c** shows the obtained recognition rates of the digitals through testing 10000 images (not overlap with the training images). After 125 training epochs, the overall accuracy rate improves from 45% to 70% with the increasing conductance margin, hinting that the higher conductance margin has a positive effect on the recognition rate for MNIST patterns. According to the simulation results, we believe the proposed InSe artificial synaptic devices are promising in the field of neuromorphic computing.

Figure R2. (a) The potentiation and depression weight states of the conductance (G) extracted from InSe A-FET device under 100 successive input pulses. The inset shows the G_{\max}/G_{\min} (conductance margin) under 50 pulses (38.1) to 100 pulses (65.3), respectively. (b) The schematic of ANN based on InSe A-FET devices and a three-layer perceptron model for image recognition. (c) The obtained recognition rates through testing 10000 images (not overlap with the training images), as a function of training phases for 100 weight states.

Besides, in this work, we systematically explored the physical mechanisms of charge trapping/detrapping behaviour in the InSe FET and developed its promising applications in nonvolatile memory and artificial synapse. Compared to the related works for the 2D materials based electronics, there are several advantages of the selected InSe FET: (1) The formed native oxide layer can not only provide controllable charge trapping states, but also lead to few impacts on the electrical properties of InSe FET; (2) The current on-off ratio and high mobility of InSe channel have the potential for designing high-speed electronics; (3) It is believed that through improving the device geometry such as channel length or high-k dielectric materials, the spiking duration time, spiking voltage, and the response current are possibly deduced to achieve the low power consumption systems. Therefore, taking into consideration of the advantages of InSe system as mentioned and the former experience of the network-level functionalities for neuromorphic computing by other van der Waals materials (graphene or MoS₂ thin films with a large-scale size), we believe a better performance based on the proposed devices could be potentially done in the near future and we will also make our best effort on it.

Therefore, based on the Reviewer #1's constructive suggestions, to make a clearer description, the sentence "Furthermore, the fitted curves based on the exponential functions located in the first and third quadrants were highly consistent with the excitatory feature in the typical biological synapse, which implied the important availability of native oxide-inspired van der Waals devices for designing promising neuromorphic architectures." has been revised as "The fitted curves based on the exponential functions located in the first and third quadrants were highly consistent with the excitatory feature in the typical biological synapse." in the revised version on **Page 13**. Besides, the simulation results of the system-level pattern recognition based on InSe artificial synapse are added in the new version of the manuscript in **Figure 5** and **Figure S12**. The related descriptions "Furthermore, we constructed a learning platform based on a three-layer perceptron network model and the above InSe artificial synaptic devices to simulate the system-level pattern recognition. As shown in Figure 5i, the designed artificial neural network (ANN) consists of 784 input neurons, 150 hidden layer neurons, and 10 output neurons. The binarized images (28×28 pixels) in the MNIST (Modified National Institute of Standards and Technology) handwritten database and the obtained digitals (0-9), serving as the image data vector and output vector, were assigned to the input and output layers, respectively. The variable weight connections among these neurons, as the synaptic weight vector, correspond to the potentiation and depression conduction states of InSe synaptic devices, which were experimentally extracted under successive input spikes (Figure 5h). To perform

the recognition tasks for digital images, 60000 images were used from the MNIST database during the learning process; the vector conversion via the sigmoid activation function and the weight update via the back-propagation learning algorithm were involved in. Note that the cycle-to-cycle weight update variation (σ) was set to 1%. Figure 5j shows the obtained recognition rates of the digitals (**Figure S12** for 50 weight states) after 125 training epochs, with each epoch size of 8000 images. The overall accuracy rate improves from 45% to 70% with the increasing conductance margin, hinting at the positive effect of higher conductance margin on the recognition rate for MNIST patterns. On the other hand, compared to the previous works for the 2D materials based electronic devices, the proposed InSe A-FET device presents three vital advantages: (1) The interfacial native charge trapping layer has little impact on electrical properties or stability; (2) The current on-off ratio and high mobility of InSe shows the potential for developing high-speed electronics; (3) The low power consumption systems can be expected through improving high-k dielectric materials or diminishing the channel length to deduce the spike duration time, spike voltage, as well as the response current. Thus we believe that the emulation of synaptic functions in device level and the simulation of pattern recognition in system-level in this work further demonstrate the important availability of native oxide-inspired van der Waals devices for designing promising neuromorphic architectures.” have been supplemented on **Page 14** for demonstrating the potential of our proposed devices in the field of neuromorphic computing. The relative references have been added as **Ref 75** (NeuroSim+: An integrated device-to-algorithm framework for benchmarking synaptic devices and array architectures. 2017 IEEE *International Electron Device Meeting 6.1.1-6.1.4*, San Francisco, CA. (2017)).

Thank Reviewer #1 again for putting up forward the worthy issues about the network-level demonstration experimentally for our future studies. We deeply hope that our explanation and effort can offer a more suitable explanation for both Reviewer #1 and the Editor.

Response to Reviewer #3:

Reviewer #3: The authors use the presence of native oxide on InSe to show synaptic behavior in InSe FETs. My major concern is in the synapse characterizations.

We thank the Reviewer #3 very much for carefully reviewing our manuscript and providing us with the professional and constructive comments, which are very important for us to further improve our work and the quality of the manuscript. In this work, we systematically explored the physical mechanisms of charge trapping/detrapping behaviour in the InSe FET and developed its promising applications in nonvolatile memory and artificial synapse. Through the similar oxidation phenomena in other layered materials were proposed by recently reported works, such as black phosphorus (*Adv. Mater.* 2016, **28**, 4991), the intrinsic charge transfer process and physical mechanisms are always mentioned lightly. Besides, compared to the related works for the 2D materials based electronics, there are several advantages of the selected InSe FET system in this work: (1) The formed native oxide layer can not only provide controllable charge trapping states (i.e. hysteresis size), but also slightly impact on the electrical properties of InSe FET (i.e. on/off current modulation, mobility, subthreshold swing and so on.....); (2) The current on-off ratio and high mobility of InSe channel show the potential for designing high-speed electronics; (3) It is believed that through improving the device configuration such as channel length or high-k dielectric materials, the spiking duration time, spiking voltage, and the response current could be deduced to achieve the low power consumption systems. We believe that developing reliable methods for analyzing and understanding the electron traps in the oxide layer is of great significance for limiting its negative impact even utilizing it for enhancing the performance and applications of electronic devices. Several sentences to emphasize the importance of our current work have been inserted into the Introduction part on **Page 3 and Page 14**. Besides, the **system-level simulation** results of an artificial neural network based on the InSe FET device for pattern recognition have been supplemented as **Figure 5 on Page 14**. **We deeply hope our effort and revised manuscript can let the readers grasp the significance of oxidation-boosted charge trapping in layered electronics, while the existence of the oxide layer is intuitively taken as a drawback, particularly for nanoscale electronics, but here can be beneficial in realizing potential applications.**

According to the valuable suggestions, we rigorously examined the experimental details and revised the manuscript carefully. For convenience, each of the comments by Reviewer #3 is answered point-by-point and all of the related corrections are highlighted in the new version of the manuscript.

Comment 1:

The authors show the variation of synaptic weight update with variable pulse width. The standard and algorithmically feasible approach is to use equal pulse width for all input pulses. How does the synaptic weight change when pulses of the same width are applied in succession?

Response 1:

The authors would like to thank Reviewer #3 for pointing out the professional comment about the evolution of the synaptic weight change under successive spike pulse with the same pulse width. In the characterizations of artificial synaptic behaviour, the synaptic weight was increased along with the increasing pulse width (as shown in Figure 5c), indicating the controllable charge trapping process in the oxide layer by different pulse duration time. Besides, as suggested by the Reviewer #3, the successive input pulses were executed with the same pulse width (Figure 5f) depending on different electrostatic amplitudes (i.e. gate voltage) to examine the potentiation and depression behavior of the InSe artificial synaptic device. Based on Reviewer #3's suggestions, the corresponding synaptic weight changes as a function of pulse number are extracted, as shown in **Figure R3**. As we can see, the synaptic weight gradually increases under 50 sequential input pulses when the gate voltage is negative, corresponding to the potentiation behaviour of the InSe artificial synapse. While the synaptic weight gradually decreases under 50 sequential input pulses when the gate voltage is positive, corresponding to the depression behaviour of the InSe artificial synapse. Therefore, to make a clear presentation, the relative results about the extracted evolution of the synaptic weights of InSe artificial synaptic devices are added as **Figure S11** in the revised version. The related description of "The extracted synaptic weight changes depending on the sequential input pulses were provided in **Figure S11**." has been added in the revised manuscript on **Page 13**.

Furthermore, we also recorded the evolution of the potentiation and depression states for InSe A-FET device under 100 successive input pulses to perform the system-level simulation of pattern recognition based on a three-layer perceptron network model. **Figure R4a** shows the corresponding changes of the conductance (G) under successive pulses with equal pulse width and the extracted conductance margins (G_{\max}/G_{\min}). **Figure R4b** illuminates the designed artificial neural network (ANN), including 784 input neurons, 150 hidden layer neurons, and 10 output neurons. In detail, the binarized images (28×28 pixels) and the obtained digitals (0-9) were assigned to 784 input neurons and 10 output neurons, respectively. The variable weight connections among these neurons correspond to the potentiation and depression conduction states of InSe synaptic devices (**Figure R4a**). During the network learning, 60000 images were used from the MNIST database; the processes involved in the vector conversion via the sigmoid activation function and the weight update via the back-propagation learning algorithm. The obtained recognition rates of the digitals through testing 10000 images (not overlap with the training images) were shown in **Figure R4c**. As the conductance margin increases from 38.1 to 65.3, the overall accuracy rate improves from 45% to 70% after 125 training epochs, which hints that the conductance margin has a positive effect on the recognition rate for MNIST patterns. Therefore, based on the obtained simulation results and the abovementioned advantages of InSe system, we believe a better performance based on the proposed devices could be potentially done in the field of neuromorphic computing in the near future and we will also make our best effort on the experimental realization in system level. For making a better demonstration, the simulation results of the system-level pattern recognition based

on InSe artificial synapse are supplemented in the new version of the manuscript as **Figure 5**. The related descriptions “Furthermore, we constructed a learning platform based on a three-layer perceptron network model and the above InSe artificial synaptic devices to simulate the system-level pattern recognition. As shown in Figure 5i, the designed artificial neural network (ANN) consists of 784 input neurons, 150 hidden layer neurons, and 10 output neurons. The binarized images (28×28 pixels) in the MNIST (Modified National Institute of Standards and Technology) handwritten database and the obtained digitals (0-9), serving as the image data vector and output vector, were assigned to the input and output layers, respectively. The variable weight connections among these neurons, as the synaptic weight vector, correspond to the potentiation and depression conduction states of InSe synaptic devices, which were experimentally extracted under successive input spikes (Figure 5h). To perform the recognition tasks for digital images, 60000 images were used from the MNIST database during the learning process; the vector conversion via the sigmoid activation function and the weight update via the back-propagation learning algorithm were involved in. Note that the cycle-to-cycle weight update variation (σ) was set to 1%. Figure 5j shows the obtained recognition rates of the digitals (**Figure S12** for 50 weight states) after 125 training epochs, with each epoch size of 8000 images. The overall accuracy rate improves from 45% to 70% with the increasing conductance margin, hinting at the positive effect of higher conductance margin on the recognition rate for MNIST patterns. On the other hand, compared to the previous works for the 2D materials based electronic devices, the proposed InSe A-FET device presents three vital advantages: (1) The interfacial native charge trapping layer has little impact on electrical properties or stability; (2) The current on-off ratio and high mobility of InSe shows the potential for developing high-speed electronics; (3) The low power consumption systems can be expected through improving high-k dielectric materials or diminishing the channel length to deduce the spike duration time, spike voltage, as well as the response current. Thus we believe that the emulation of synaptic functions in device level and the simulation of pattern recognition in system-level in this work further demonstrate the important availability of native oxide-inspired van der Waals devices for designing promising neuromorphic architectures.” have been added on **Page 14**. Thanks very much for the reviewer to raising these constructive comments. The authors hope that our effort can offer a suitable explanation to Reviewer #3.

Figure R3. (a) The evolution of the calculated synaptic weight (A_n/A_0) depending on the number of successive input pulses. Note that the input gate voltages are varied from -20 to -60 V. (b) The corresponding synaptic weight under the stimulation of successive positive voltage pulses varying from 20 to 60 V.

Figure R4. (a) The potentiation and depression weight states of the conductance (G) extracted from InSe A-FET device under 100 successive input pulses. The inset shows the G_{max}/G_{min} (conductance margin) under 50 pulses (38.1) to 100 pulses (65.3), respectively. (b) The schematic of ANN based on InSe A-FET devices and a three-layer perceptron model for image recognition. (c) The obtained recognition rates through testing 10000 images (not overlap with the training images), as a function of training phases for 100 weight states.

Comment 2:

In Fig. 5b, what are the various colors signifying? I assume those are for different pulse amplitudes? It needs to be clear. What are those amplitudes? Please put the label on the plot or in the caption.

Response 2:

The authors would like to thank Reviewer #3's constructive suggestions about the description of the measurement details for mimicking the synaptic behavior of InSe A-FET. In our manuscript, as Reviewer #3 assumed, Figure 5b shows the obtained postsynaptic current responses under different gate voltage pulses with various amplitudes, which were marked by different colors. The values of the voltage amplitudes, from the top curve to the bottom one, are -80 , -70 , -60 , -50 , and -40 V,

respectively. Based on the Reviewer #3's suggestions, to make a clearer description, the corresponding gate voltage values are marked in the **Figure 5b** (as shown in **Figure R5b**) and are also highlighted in the caption of **Figure 5b** as “EPSC generated by applying several input spikes with different voltage amplitudes under the potentiation condition of -80 , -70 , -60 , -50 , and -40 V, respectively.” The authors hope that our effort can offer a clearer presentation to Reviewer #3.

Figure R5. (a) Schematic illustration of a biological synapse and sketch of the InSe artificial synaptic device. Note that the gate electrode serves as the presynaptic terminal and the current in the InSe channel is considered as the PSC response. (Courtesy of <https://tw.123rf.com> for creative commons attribution.) (b) EPSC generated by

applying several input spikes with different voltage amplitudes under the potentiation condition of -80 , -70 , -60 , -50 , and -40 V, respectively. The drain voltage is set as 0.1 V under the Read operations in this work. The change of PSC after the input spike (ΔPSC) is labeled by arrows. Inset shows the calculated values of w for the case of InSe A- (Brown) and V-FET (Turquoise) devices. (c) Variation of w under different pulse widths. (d) Extracted PPF index $((A_2 - A_1)/A_1)$ as a function of spike time interval Δt , where A_1 and A_2 are the first and second EPSC peak, respectively. The solid line is the fitted curve based on the double exponential function. (e) The plot of IPSC changes over 120 s after stimulating by various numbers of the input pulse (pulse number= 1, 5, 10, 30, and 50, respectively). For a better comparison, inset shows the corresponding changes after 1 and 50 pulses in the case of InSe V-FET. (f) Monitored PSC as a function of pulse number with varying amplitudes of sequential voltage pulses (pulse width of 50 ms, time interval of 100 ms). The application of the positive and negative voltage pulse sequences resulted in dynamic depression and potentiation behaviours in PSC. (g) Changes of synaptic weight (Δw) depending on the Time interval (ΔT) of the pre- and postsynaptic simulations and the corresponding fitted curves based on the exponential functions, which demonstrates STDP behaviour in the InSe artificial synaptic device. Note that the Δw is defined as $(\Delta\text{PSC}_2 - \Delta\text{PSC}_1)/\Delta\text{PSC}_1$, where ΔPSC_1 and ΔPSC_2 represent the obtained changes of PSC response after the pre- and postsynaptic simulations, respectively. The fitted lines are provided as a guide to the eye. The inset shows the schematic of the separated input signals from two terminals. (h) The potentiation and depression weight states of the conductance (G) extracted from InSe A-FET device under 100 successive input pulses. The inset shows the $G_{\text{max}}/G_{\text{min}}$ (conductance margin) under 50 pulses (38.1) to 100 pulses (65.3), respectively. (i) The schematic of ANN based on InSe A-FET devices and a three-layer perceptron model for image recognition. (j) The obtained recognition rates through testing 10000 images (not overlap with the training images), as a function of training phases for different weight states and conductance margins.

Comment 3:

Same comment as above for Figure 5e.

Response 3:

The authors would like to thank Reviewer #3's constructive suggestions about the presentation of the measurement details. As we can see, **Figure R5e** shows the retention of the read current over 120 s after stimulating by different numbers of input spikes, which indicates the long-term plasticity of the InSe artificial synaptic device. Note that the five curves with different colors respond to the pulse number= 1, 5, 10, 30, and 50, respectively. Based on the Reviewer #3's suggestions, to make a clearer description, the corresponding information has been added in the caption of **Figure 5e** as "The plot of IPSC changes over 120 s after stimulating by various numbers of the input pulse (pulse number= 1, 5, 10, 30, and 50, respectively)." Thanks sincerely for the reviewer to point out the constructive suggestions and hope that our effort can offer a clearer presentation to Reviewer #3.

Comment 4:

What is the pulsing scheme for STDP of Fig. 5g? Please show.

Response 4:

The authors would like to thank Reviewer #3 for pointing out the professional comments about the pulse scheme for STDP in the characterizations of artificial synapse features. In the biological synapse, STDP is of importance to the Hebbian synaptic learning and memory functions, which relates to the time interval and the spike order of the pre- and postsynaptic stimulations. In our work, the STDP of the artificial synaptic device was performed by applying two separated input signals from the back gate terminal and the drain terminal, respectively. The obtained results of the potentiation ($\Delta T > 0$) and depression ($\Delta T < 0$) responses were shown in Figure 5g. Based on Reviewer #3's suggestions, for making a clearer presentation, the corresponding schematic of the action-potential like signals is provided in Figure R6 and added in the inset of Figure 5g (as shown in Figure R5b). As we can see, the presynaptic signal is composed of a bigger positive voltage pulse (+40 V, 100 ms) and a decreased tail (-30 V, 100 ms) because the thickness of the SiO₂ dielectric layer in our present device configuration is 300 nm; the postsynaptic signal contains a smaller positive voltage pulse (+5 V, 200 ms) for better symmetry of the STDP. The relative description is added in the caption of Figure 5g as "The inset shows the schematic of the separated input signals from two terminals." Thanks sincerely for the reviewer to point out the constructive suggestions and hope that our effort can offer a clearer presentation to Reviewer #3.

Figure R6. The schematic of the separated input signals of pre-voltage and the post-voltage from the gate terminal and the drain terminal, respectively. Note that the pulse width and the time interval (ΔT) have been marked.

Comment 5:

Minor comment: Line 40-41: Should it not be molybdenum ditelluride?

Response 5:

The authors would like to thank Reviewer #3 for pointing out this spelling lapse in time about the full name of the materials. This is a very valuable comment for us to improve the quality of the current manuscript. The words "molybdenum ditelluride (MoTe₂)" in the part of the Introduction have been revised in the new version of the manuscript.

Thank you in advance for your time and kind consideration of our manuscript.

Sincerely Yours,

Dr. Yen-Fu Lin,

Principal Investigator of Nano Research Group,

National Chung Hsing University (NCHU), Taiwan

2019 MOST Ta-You Wu Memorial Award Winner

Special commentator in *Nature Electronics*

Editorial Board Member in *Scientific Reports*

Editorial Board Member in *Physics Bimonthly*

Associate Professor, Department of Physics and Institute of Nanoscience, NCHU

Jointly Appointed Associate Professor, Research Center for Sustainable Energy and Nanotechnology,
NCHU

Website: <http://linyf.nchu.edu.tw/>

E-mail: yenfulin@nchu.edu.tw

REVIEWERS' COMMENTS:

Reviewer #1 (Remarks to the Author):

The authors have addressed my previous concerns appropriately, so that I am satisfied with the revision that the authors have made. I recommend for acceptance.

Reviewer #3 (Remarks to the Author):

The authors have taken great efforts in answering my concerns. I am satisfied and I recommend this paper for publication at this stage.